# Structural Repetition Detector for multi-scale quantitative mapping of molecular complexes through microscopy

Afonso Mendes[1,2,3], Bruno M. Saraiva [1,2,3], Guillaume Jacquemet [4,5,6,7], João I. Mamede [8], Christophe Leterrier [9] ✉ & Ricardo Henriques [1,3,10] ✉

From molecules to organelles, cells exhibit recurring structural motifs across multiple scales. Understanding these structures provides insights into their functional roles. While super-resolution microscopy can visualise such patterns, manual detection in large datasets is challenging and biased. We present the Structural Repetition Detector (SReD), an unsupervised computational framework that identifies repetitive biological structures by exploiting local texture repetition. SReD formulates structure detection as a similarity-matching problem between local image regions. It detects recurring patterns without prior knowledge or constraints on the imaging modality. We demonstrate SReD's capabilities on various fluorescence microscopy images. Quantitative analyses of different datasets highlight SReD's utility: estimating the periodicity of spectrin rings in neurons, detecting Human Immunodeficiency Virus type-1 viral assembly, and evaluating microtubule dynamics modulated by End-binding protein 3. Our open-source plugin for ImageJ or FIJI enables unbiased analysis of repetitive structures across imaging modalities in diverse biological contexts.

Biological systems exhibit structural repetition across multiple scales, from biomolecules to supramolecular assemblies and cellular structures[1]. Understanding these patterns is crucial for identifying their functional significance and underlying biological processes[2]. Microscopy techniques offer molecular-level resolution but manually detecting repetitive motifs in large datasets is impractical, biased, and expertise-dependent[3]. To address these limitations, machine learning, particularly deep convolutional neural networks (CNNs), has been employed to detect and segment biological structures automatically[4]. However, CNNs require extensive labelled training data, inheriting biases[5]. Previous

methods enable unbiased registration but need point data, limiting their applicability[6,7]. We present the Structural Repetition Detector (SReD), an unsupervised framework to identify repetitive biological structures by exploring local texture redundancy. SReD formulates structure detection as similarity matching between local image regions, allowing pattern detection without prior knowledge or microscopy modality constraints. We demonstrate SReD's capabilities on fluorescence microscopy images of diverse cell types and structures, including microtubule networks, nuclear envelope, pores, and virus particles (Fig. 1). SReD generates Structural Repetition Scores (SRSs) highlighting regions with repetitive

[1]Optical Cell Biology group, Instituto Gulbenkian de Ciência, Oeiras, Portugal. [2]Gulbenkian Institute for Molecular Medicine, Oeiras, Portugal. [3]Instituto de Tecnologia Química e Biológica António Xavier, Universidade Nova de Lisboa, Oeiras, Portugal. [4]Turku Bioimaging, University of Turku and Åbo Akademi University, Turku, Finland. [5]Faculty of Science and Engineering, Cell Biology, Åbo Akademi University, Turku, Finland. [6]InFLAMES Research Flagship Center, Åbo Akademi University, Turku, Finland. [7]Turku Bioscience Centre, University of Turku and Åbo Akademi University, Turku, Finland. [8]Department of Microbial Pathogens and Immunity, Rush University Medical Center, Chicago, IL, USA. [9]Aix Marseille Université, CNRS, INP UMR7051, NeuroCyto, Marseille, France. [10]UCL Laboratory for Molecular Cell Biology, University College London, London, United Kingdom. ✉e-mail: christophe.leterrier@univ-amu.fr; r.henriques@itqb.unl.pt

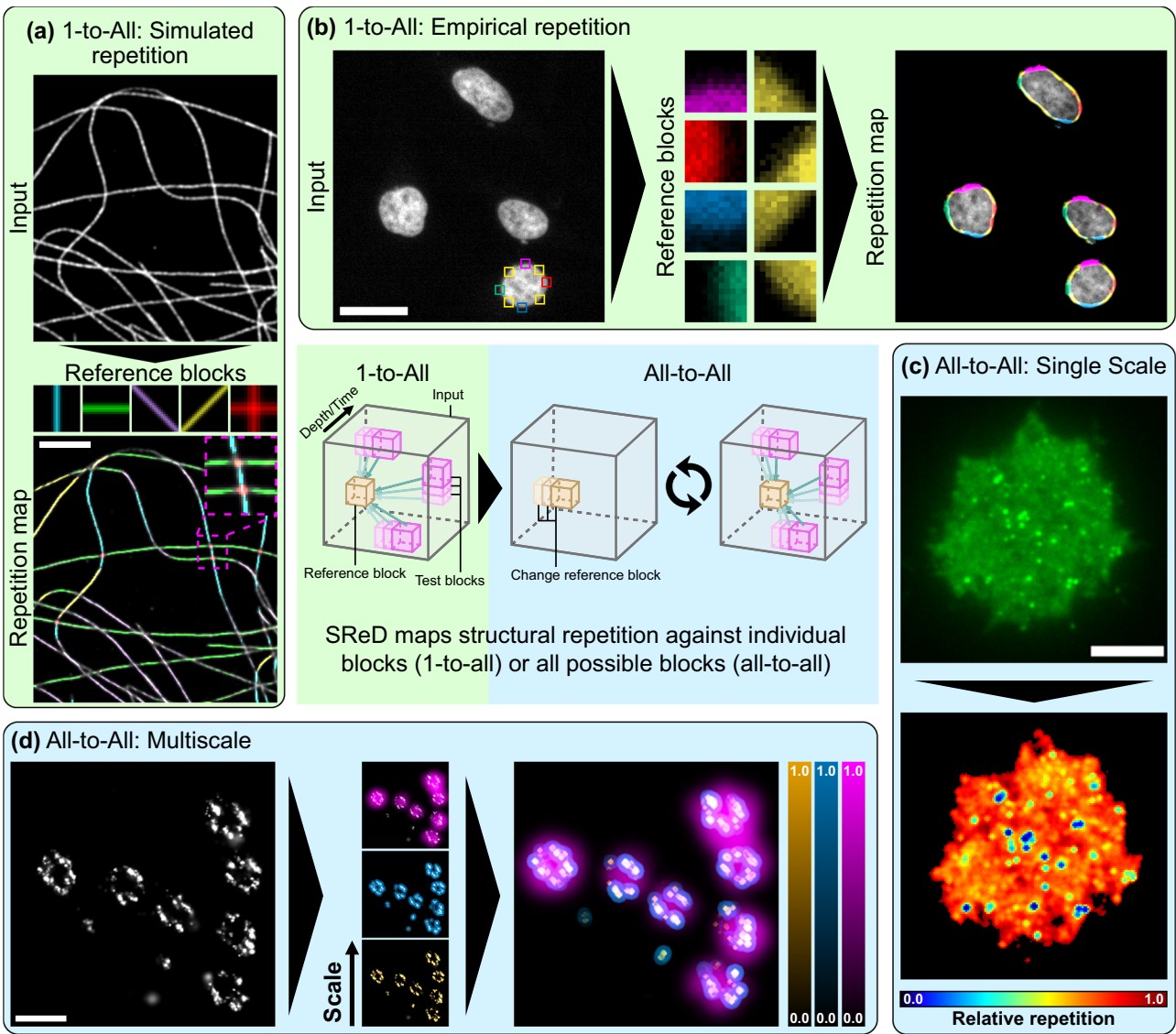

**Fig. 1 | Applications of the Structural Repetition Detector (SReD) algorithm in fluorescence microscopy. a** Detection of Structural Repetition Using Simulated Blocks: Microtubules imaged with STORM analysed for repetitive patterns using simulated structural blocks. The analysis was performed using SReD's 'block repetition' mode, which features a '1-to-all' matching scheme where a reference block is compared with all the remaining image blocks. A rotation-variant correlation metric (Pearson's correlation coefficient) was used to account for structure orientation. Coloured regions in the repetition map correspond to repetitions of same-coloured blocks above. Scale bar: 2 μm. **b** Detection of Structural Repetition Using Empirical Blocks: HeLa cell nuclei stained with DAPI used to detect repetitive structural patterns using manually extracted empirical reference blocks. The analysis was performed in a similar manner as shown in (**a**) but the reference blocks are extracted directly from the input data. Coloured regions in the repetition map correspond to repetitions of same-coloured blocks in the previous subpanel. Scale bar: 30 μm. **c** Global Repetition Detection: Jurkat cell expressing inducible HIV Gag-

EGFP fusion protein analysed using global repetition detection and a rotation-invariant metric ('absolute difference of standard deviations'). The analysis was performed using SReD's 'global repetition' mode, which features an 'all-to-all' matching scheme where all image blocks are compared with all the remaining image blocks. The repetition map reveals structures not easily detectable in input image and their relative frequency. Scale bar: 5 μm. **d** Multiscale Global Repetition: *Xenopus laevis* nuclear pores imaged with STORM analysed using different-sized receptive fields to detect structural repetition at various scales. The analysis was performed using SReD's 'global repetition' mode, where each iteration used a different block-to-image size ratio. The repetition map identifies repeated structures from single nucleoporins (orange) to nucleoporin clusters (blue) and nuclear pore units (magenta). A rotation-invariant correlation metric was used to analyse structures irrespective of their orientation. Scale bar: 120 nm. Centre panel: Simplified SReD algorithm workflow, illustrating key steps from input preprocessing to repetition map generation.

textures. Users can provide artificial blocks or extract them from the data for repetition analysis. An unbiased sampling scheme maps global repetition by testing every possible image block as a reference (Supplementary Note 1). We showcase SReD's utility through three datasets: 1) spectrin rings in neuronal axons, accurately estimating ring periodicity and pinpointing periodic patterns, 2) Human Immunodeficiency Virus (HIV) Gag assembly, mapping viral structures without structural priors, and 3) dynamic

End-Binding Protein 3 (EB3) and microtubule structures, assessing structural displacement and stability over time. Our open-source ImageJ/FIJI[8,9] plugin enables versatile, unbiased analysis of redundancy in microscopy images. SReD advances computational microscopy by providing a generalised framework for detecting repetitive structures without labelled training data or single-molecule localisation input, facilitating the quantitative study of structural motifs across scales in diverse imaging datasets.

## Results

### Implementation, theoretical foundation and core functionality

SReD is an open-source ImageJ and Fiji[8,9] plugin that leverages graphics-processing unit (GPU) acceleration to identify repetitive patterns in microscopy images. The algorithm's workflow, outlined in Fig. 1 (centre panel), begins with the application of the Generalised Anscombe Transform (GAT) to stabilise noise variance (Supplementary Note 1)[10] This step addresses the noise in microscopy images, which often exhibit Poisson and Gaussian noise. The GAT nonlinearly remaps pixel values to produce an image with near-Gaussian noise and stabilised variance, preserving local contrast and overall image statistics. This stabilisation is essential for robust downstream processing, mitigating violations of normality, homoscedasticity, and outlier assumptions that can compromise correlation metrics. Following noise stabilisation, SReD generates a relevance mask to exclude regions lacking substantive structural information, based on local texture prominence quantified by variance (Supplementary Note 1; Supplementary Fig. 1). The rationale is that structural elements present themselves as regional image textures with non-zero variance. Therefore, areas devoid of structure will exhibit minimal texture. Due to the ubiquitous presence of noise, we calculate a threshold at which image texture is minimal by estimating the average noise variance[11]. The final relevance threshold is defined by multiplying the estimated average noise variance by an adjustable constant, with the default set at 0. This produces a binary mask outlining areas with sufficient structural content. The analysis proceeds using reference blocks, either simulated or sampled from the image (i.e., empirical). These blocks are matched against the input using correlation metrics to generate repetition maps (Supplementary Note 1). Our algorithm leverages a custom sampling scheme in which a reference block is compared with all possible test blocks in the image. The scheme can be '1-to-all' or 'all-to-all', depending on the application. The first requires a user-provided reference block, while the latter provides unbiased structure detection. The comparisons between blocks consist of calculating correlation metrics. The correlation metrics can be rotation-variant (e.g., Pearson's correlation coefficient) or -invariant (e.g., absolute difference of standard deviations (ADSD))(Supplementary Note 1). In both cases, the blocks' dimensions are predefined by the user to match a specific scale. A repetition map is calculated for each '1-to-all' comparison, where each pixel is assigned a score (named Structural Repetition Score, or SRS), which reflects the similarity between the local neighbourhood centred at that position and the reference block. Finally, the repetition map is normalised to its range. The SRS is given by Eq. 1.

$$SRS\left(X_i, Y_j\right) = Corr\left(X_i, Y_j\right) \cdot Rel\left(Y_j\right) \tag{1}$$

where $X_i = \{x_1, x_2, \ldots, x_n\}$ and $Y_j = \{y_1, y_2, \ldots, y_n\}$ are the reference and test blocks with size $n$ (in pixels) centred at pixel positions $i$ and $j$, and

$$Rel\left(Y_j\right) = \begin{cases} 0, & if\ Var(Yj) \leq \overline{Var} \\ 1, & if\ Var(Yj) > \overline{Var} \end{cases} \tag{2}$$

the binary 'relevance' label of the test block is calculated as in Eq. 2, where $\overline{Var}$ is the average noise variance of the input image. To analyse local textures and calculate a single value for each, the reference and test blocks require a defined centre. Therefore, the blocks' dimensions need to be odd, and as a result, $i = [(r_h, H - r_h]$ and $j = [r_w, W - r_w]$, where $r_w$ and $r_h$ are the blocks' width and height radii, and $W$ and $H$ are the input image's width and height. In the block repetition mode, the input image is probed for repetitions of a single reference block using the '1-to-all' sampling scheme. This generates a repetition map reflecting the likelihood of the reference pattern occurring at that each location. In

the global repetition mode, SReD enables unbiased structure analysis by using the entire universe of image blocks as a reference ('all-to-all' sampling scheme). Each reference block generates a repetition map that is averaged, and the average value is plotted at the coordinates corresponding to the centre of the reference block. The average uses an exponential weight function based on the distance between the standard deviations of the blocks in each comparison, which enhances structural details. Therefore, the global repetition scores represent the relative repetition of a local texture across the image. Mathematically, the global SRS is calculated as in Eq. 3.

$$GSRS\left(X_i, Y_j\right) = \frac{\sum_j Corr\left(X_i, Y_j\right) \cdot W\left(X_i, Y_j\right) \cdot Rel\left(Y_j\right)}{N \cdot \sum_j W\left(X_i, Y_j\right)} \tag{3}$$

where $N$ is the size of the input image (excluding borders with length equal to the XY radii of the blocks). The exponential weight function is defined in Eq. 4.

$$W\left(X_i, Y_j\right) = e^{-\left|\frac{\sigma_{X_i} - \sigma_{Y_j}}{\overline{Var}}\right|^2} \tag{4}$$

Here, in Eq. 4., $\sigma_{Xi}$ and $\sigma_{Yj}$ are the standard deviations of the reference and test blocks. The resulting repetition maps highlight regions likely to contain structural repetitions. Non-linear mapping can be applied to enhance the contrast between different SRSs within the repetition maps, facilitating visual interpretation and subsequent analysis. In our experience, we have found that applying a power transformation to the SRSs often yields the most effective enhancement. This transformation involves raising each SRS value to a specific exponent. The choice of exponent plays a crucial role in determining the degree of contrast enhancement. In this study, we explored a range of exponents between 10 and 10,000. Typically, we initiate the analysis with an exponent of 10 and iteratively adjust it based on the visual assessment of the resulting repetition map. For datasets with subtle structural repetitions or low signal-to-noise ratios, higher exponents may be necessary to amplify the differences between SRSs and reveal hidden patterns. Conversely, for datasets with prominent structural repetitions, lower exponents may suffice to achieve adequate contrast enhancement without introducing excessive noise amplification. The optimal exponent ultimately depends on the specific characteristics of the data and the desired level of visual clarity. By carefully selecting the exponent, users can tailor the contrast enhancement to their needs, facilitating the identification and interpretation of repetitive patterns in diverse microscopy images.

### General applications of SReD

To demonstrate SReD's versatility across diverse biological contexts, we conducted a comprehensive analysis of various microscopy datasets (Fig. 1; Supplementary Note 2). We first examined a Stochastic Optical Reconstruction Microscopy (STORM) image reconstruction of a cell with labelled microtubules[12] using the Pearson's correlation coefficient as a rotation-variant correlation metric. This approach effectively mapped microtubules at various orientations and crossings (Fig. 1a; Supplementary Fig. 2). We further illustrate SReD's versatility by detecting nuclear envelopes in DAPI-stained cells[13] following the same approach but using empirical reference blocks extracted directly from the input image. Here, SReD distinguished different morphological states potentially related to cell division or stress (Fig. 1b; Supplementary Fig. 3; Supplementary Note 2). SReD also enables characterisation of structures without user-provided references. We exemplify this functionality by analysing an image of a Jurkat cell expressing an HIV Gag-EGFP construct, which induces the production of virus-like particles (VLPs) (Fig. 1c; Supplementary Fig. 4). In this mode, SReD mapped every structure in the image and assigned scores

based on their relative repetition. A rotation-invariant correlation metric (ADSD) was used to analyse structures irrespective of their orientation. As expected, the top score was given to the most repeated element, the diffuse EGFP signal, with viral structures exhibiting lower frequencies. Localisation of round viral structures via local extrema calculation revealed that the repetition map provided a superior platform for extrema detection compared to direct analysis of the raw images (Supplementary Fig. 4). The algorithm's multiscale analysis capability is achieved by adjusting the ratio of block-to-image dimensions. Larger ratios capture larger structures, while smaller ratios capture finer details. For computational efficiency, it is preferable to modulate scale by downscaling the input rather than enlarging blocks, although combining both approaches often preserves structural detail best. We demonstrate this multiscale analysis by examining nuclear pore complexes in STORM image reconstructions with labelled gp210 proteins[14] also using the ADSD metric. SReD successfully mapped structures across different scales, discerning single nucleoporins, nucleoporin clusters, and entire nuclear pores (Fig. 1d; Supplementary Fig. 5). SReD's utilisation of image reconstructions and intensity-based analysis ensures broad applicability across all types of microscopy data. We demonstrate this versatility by detecting and classifying HIV viral particles in transmission electron microscopy (TEM) data, achieving a 90% concordance compared to a CNN-based approach[15] (Supplementary Note 3; Supplementary Fig. 6; Supplementary Movie 1). Furthermore, SReD enabled the detection and quantitative description of HIV assembly platforms using STORM data, allowing for an assessment of how actin-debranching drug CK666 influences the stability of these platforms (Supplementary Note 4; Supplementary Fig. 7). A comparative analysis of SReD's capabilities against other methodologies in the same domain is presented in Supplementary Table 1.

## 1-to-all case example: detection of spectrin ring periodicity in axons

We used SReD's block repetition mode to map and quantify the membrane-associated periodic scaffold (MPS) architecture in neuronal axons automatically and without bias (Fig. 2). The MPS, composed of actin, spectrin, and associated proteins, forms a crucial structural component of neuronal axons[16,17] Super-resolution microscopy has shown that the MPS consists of ring-like structures spaced 180–190 nm apart, with alternating actin/adducin and spectrin rings orthogonal to the axon's long axis[18] Mapping this nanoscale organisation across entire neuron samples has been challenging due to the need for manual region selection, potentially introducing bias. We analysed datasets from Vassilopoulos et al.[19]. comparing neurons treated with DMSO (control) or swinholide A (SWIN, an actin-disrupting drug). Using SReD with the rotation-sensitive Pearson's correlation coefficient metric, we developed an automated workflow to determine axon orientations by probing skeletonised neuron images with simulated lines at varying angles (Supplementary Note 5; Supplementary Fig. 8). This enabled consistent alignment of axon segments for downstream analysis. We optimised parameters for simulated ring blocks to match observed ring patterns in control data, yielding an inter-ring spacing of 192 nm, consistent with previous studies (Supplementary Fig. 9)[18,19] SReD-generated repetition maps highlighting regions of high local similarity across neuron samples, allowing automatic extraction and quantification of MPS organisation without manual region selection (Fig. 2b; Supplementary Fig. 10). We measured an average spacing of 178 nm under control conditions using autocorrelation functions of the automatically extracted periodic regions (Fig. 2c; Supplementary Fig. 11d). In agreement with Vassilopoulos et al.[19]. repetition maps showed that swinholide A treatment disrupted MPS structure, with reduced pattern prominence and frequency compared to controls (Fig. 2b, c). We used correlation metrics that minimise information loss while being aware of potential imprinting. Nonlinear mapping (e.g.,

power functions) of the output data effectively distinguishes real patterns from imprinted ones (Supplementary Figs. 2d, e, 11c). Our method accounts for neuron thickness variability and provides the average distance between patterns for additional biological insights. SReD's local repetition scores quantified the fraction of structures with MPS patterns, revealing a 39% reduction in axons with detectable periodic scaffolds after swinholide A treatment ($P < 0.001$, Fig. 2d). SReD's maps identified drug-affected regions with confidence values, offering a detailed platform for analysing structural dysregulation (Fig. 2c). SReD also showed higher statistical sensitivity, detecting a 12% reduction in pattern prominence post-treatment ($P < 0.05$) previously unreported (Supplementary Fig. 11e). To test SReD's noise robustness, we conducted a sensitivity analysis with images at varying signal-to-noise ratios (SNRs) (Supplementary Fig. 12). SReD consistently detected ring structures even at low SNRs near 1, where patterns were visually indiscernible. SReD-generated maps outperformed direct STORM reconstructions in autocorrelation analysis, reliably identifying an average inter-ring spacing of 192 nm across all SNRs, demonstrating the algorithm's robustness in detecting structural periodicity despite significant noise. We assessed SReD's specificity and robustness to pattern deformations by applying stretch deformations to test images (Supplementary Fig. 13). As the stretch factor increased, the average SRS decreased, indicating pattern disruption. However, SReD remained specific to the original pattern within the expected interval. Even at higher stretch factors, non-specific patterns were quantitatively discernible and reflected the intrinsic properties of the test data. This robustness is valuable for analysing periodic structures in diverse biological contexts, where deviations from ideal patterns are common due to sample preparation artefacts, imaging noise, or biological variability.

## All-to-all 3D case example: detecting HIV Gag assembly in 3D

The establishment of a viral infection is the product of complex host-pathogen interactions, comprising an evolutionary 'tug-of-war' where cells evolve protective mechanisms whilst viruses evolve to circumvent them. Viruses typically hijack cellular transcription and translation machinery to produce viral progeny required for viral replication[20] Therefore, viral assembly represents a critical platform for host-pathogen interactions that significantly impact infection outcomes. The HIV *gag* gene encodes the Gag polypeptide precursor, which is cleaved into several key structural components. This polypeptide aggregates at the membrane of infected cells and induces the budding of membranous viral particles[20] Expression of Gag alone is sufficient to induce the formation of non-infectious virus-like particles (VLPs)[21,22] To map viral structures in an unbiased manner, we examined an image of a Jurkat cell expressing an inducible HIV Gag-EGFP construct using SReD with the rotation-invariant ADSD metric (Fig. 3a, b). The 3D data comprised 2D images acquired with a 0.5 μm offset in the Z-axis, which enabled using SReD's 3D mode. To evaluate the algorithm's accuracy, we generated a population of simulated diffraction-limited particles with randomly distributed intensities across the image's dynamic range, which served as a reference with a known ground-truth. The ImageJ/Fiji[8,9] '3D Maxima Finder' plugin, which computes local maxima in 3D space, was used to calculate 3D local maxima corresponding to active viral assembly sites from both the input image and the repetition map (Fig. 3c). Remarkably, SReD enabled the detection of 96% of the simulated particles, compared to only 32% in the input image, demonstrating the algorithm's superior accuracy over direct analysis of input images (Fig. 3d). Visual inspection of the detected EGFP intensity signal vs. the SRS for the same pixel location revealed that high-SRS regions corresponded to input regions with a wide range of intensity values. We observed that most structures of interest were allocated to the sample fraction above an arbitrary threshold of SRS 0.8, whilst the fraction below this threshold contained mostly

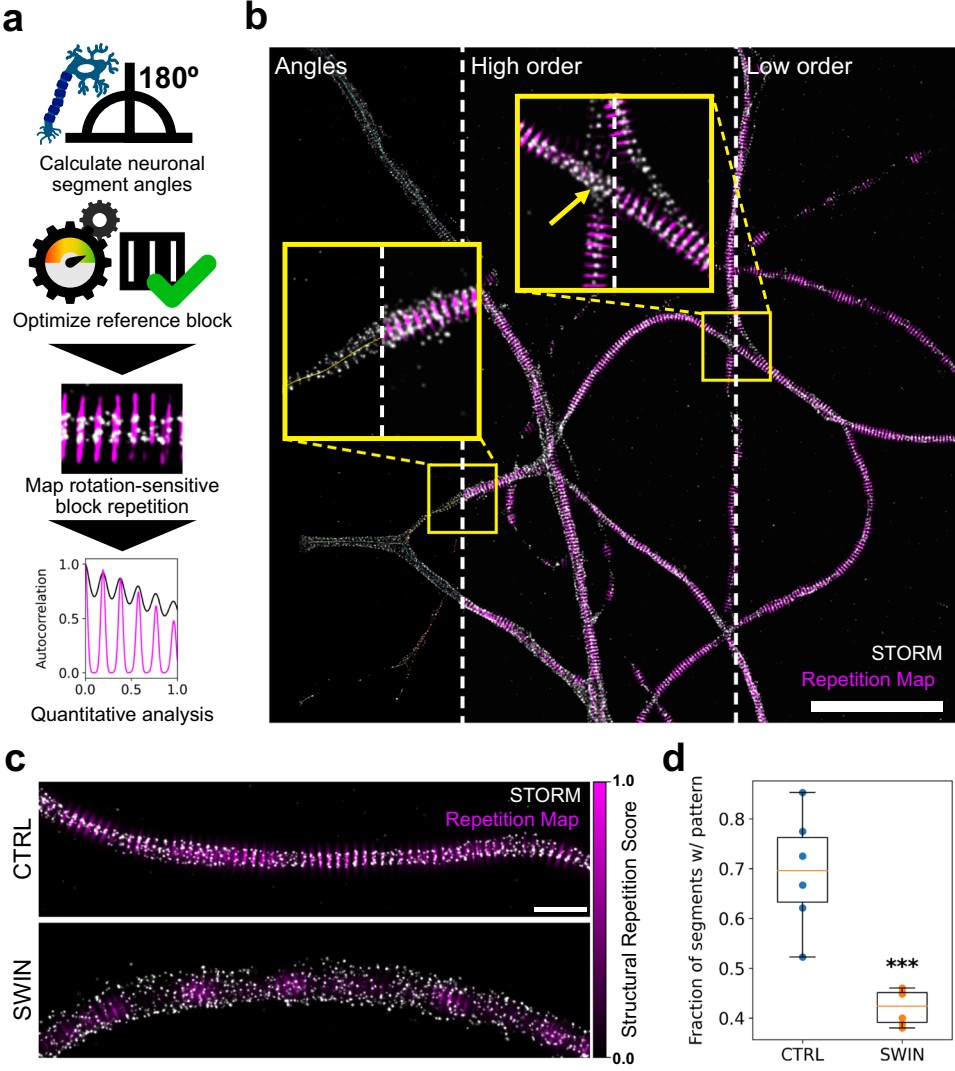

**Fig. 2 | Automated detection and quantification of spectrin ring periodicity in neuronal axons. a** SReD-based analysis pipeline. Calculate neuronal segment angles: SReD is used to determine axon orientations by detecting repetitions of reference blocks containing lines at different orientations in 'skeletonized' axons. Optimise reference block: An optimised reference block resembling a periodic ring pattern block is generated by designing test blocks with different combinations of inter-ring spacing and ring height and using SReD to determine the degree of repetition of each test block in several test input images. Autocorrelation functions are calculated for each repetition map, and the first harmonics' amplitudes are used as a measure of how well each test block fits the test data's periodic patterns. The set of parameter values with the highest average amplitude is chosen. Map rotation-sensitive block repetition: Block repetition analysis is used to find repetitions of the optimized block at various orientations. Quantitative analysis: The repetition maps are analysed by calculating autocorrelation functions and determining the average inter-ring spacing and pattern prominence from the autocorrelations' first harmonics' positions and amplitudes. All repetition analyses performed using the Pearson's correlation coefficient metric. **b** Control (CTRL) dataset image: STORM localization density (grey) overlaid with SReD repetition maps (magenta). Insets: (i)'Angles' - axon skeletons colour-coded by orientation; (ii)'High order' - repetition map calculated using a 9-ring reference block; (iii)'Low order'−repetition map calculated using a 3-ring reference block. Scale bar: 5 μm. **c** Repetition maps comparing CTRL and swinholide A-treated (SWIN) groups. SWIN-treated samples show reduced periodic structures. Scale bar: 1 μm. **d** Quantification of axon segments with ring patterns. Box plots show a significant reduction in pattern-containing segments in SWIN vs. CTRL (*n* = 6 per group, mean ± SEM; CTRL: 0.694 ± 0.008, SWIN: 0.421 ± 0.007; *p* < 0.001, two-sided unpaired *t*-test). Box plot elements are represented as such: center line as the median (50th percentile); box limits as upper (75th) and lower (25th) quartiles; whiskers indicate minimum and maximum values. Source data are provided as a Source Data file.

background signal and some reference particles (Fig. 3e). Given that autofluorescence often corrupts microscopy analyses, we evaluated the algorithm's performance in the presence of synthetic non-specific structures (Supplementary Note 6). The repetition maps produced by SReD consistently provided a superior platform for detecting simulated reference particles and viral structures across conditions (Supplementary Fig. 14). This analysis demonstrates SReD's robust capability to map biological structures, such as assembling viral particles. The algorithm's high sensitivity and specificity, even in the presence of non-specific structures, highlight its potential for studying dynamic cellular processes like viral assembly,

where the ability to accurately detect and characterise structures amidst variable backgrounds is showcased.

### All-to-all live-cell case example: Assessment of the microtubule network's stability along time

The multidimensional capabilities of SReD can be extended to analyse structural dynamics over time, providing insights into structural stability. We demonstrate this application using time-lapse imaging of RPE1 cells stably expressing EB3 fused to GFP (Fig. 4a). EB3 binds to the plus ends of microtubules, appearing as comet-like structures that travel along the cytoplasm when visualised under fluorescence

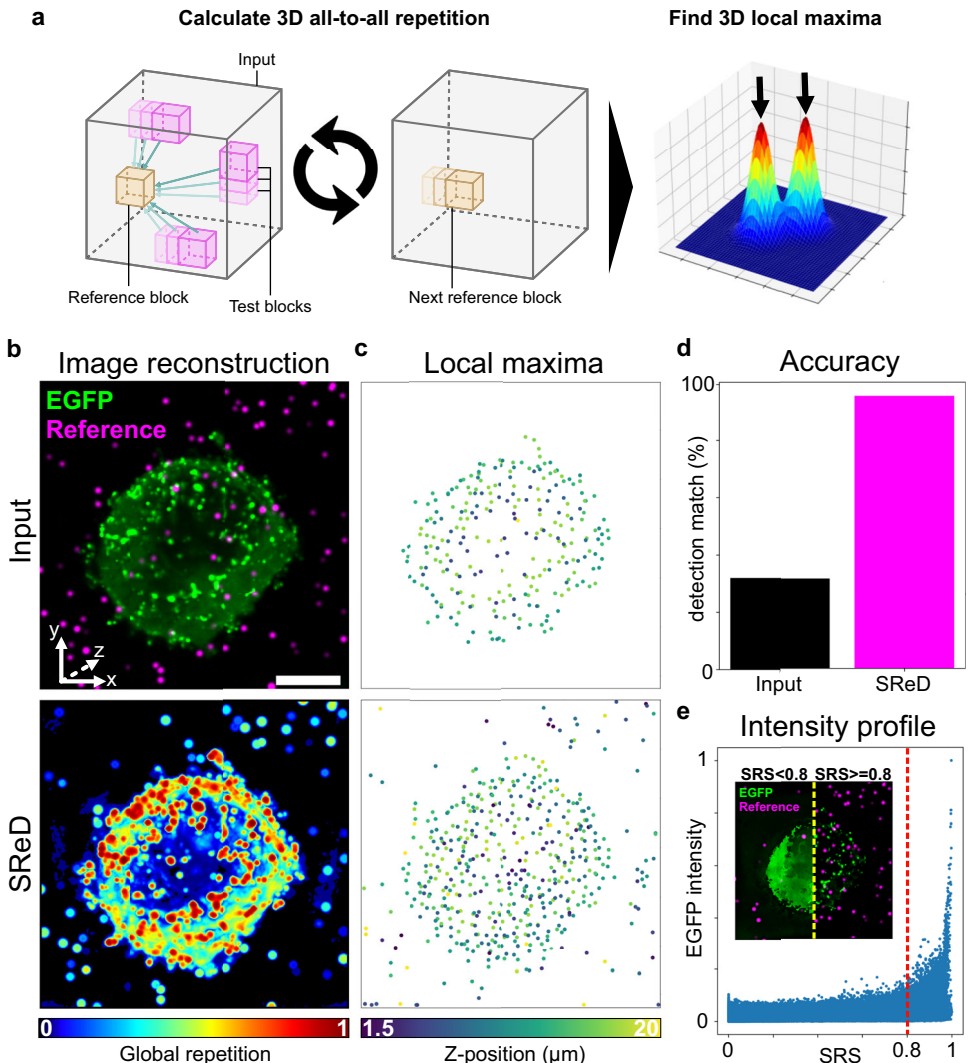

**Fig. 3 | Detecting HIV virus-like particles in 3D. a** Analysis pipeline schematic. The algorithm uses 3D reference blocks for rotation-invariant structural ('absolute difference of standard deviations metric) repetition analysis, locating viral structures via 3D local maxima detection from the input image and repetition map. **b** Z-projections of the input image (top) and repetition map (bottom), highlighting viral-like particles ('EGFP') and simulated reference particles ('Reference'). The 'Reference' particles were designed as diffraction-limited particles with randomly distributed intensities across the input's dynamic range and serve as a reference with a known ground-truth to evaluate the algorithm's accuracy. **c** Local maxima plots showing detected structures in the input image (top) and repetition map (bottom), with increased sensitivity in the repetition map. **d** Accuracy plot comparing artificially-added reference particle detection: input image (32%) vs. repetition map (96%). **e** Intensity profile graph of EGFP signal (green) and structural repetition score (SRS, magenta), with a threshold at SRS 0.8 (dashed red line). Inset shows pixels below (left) and above (right) the threshold, indicating high-SRS structures. Scale bars: 5 μm (main images), 1 μm (inset). Source data for (**c**–**e**) are provided as a Source Data file.

microscopy[23] We used SReD with the rotation-insensitive ADSD metric to generate a global repetition map by treating time as the third dimension in our analysis, using a time-lapse sequence of ~2 min (Fig. 4b). To quantify structural changes, we calculated the Normalised Root Mean Squared Error (NRMSE) between the first and last frames of the time-lapse for both the input images and the repetition maps. The NRMSE of the input images reflected the spatial displacement of dynamic structures, yielding a relatively low value. In contrast, the NRMSE calculated from the repetition maps was higher, indicating greater sensitivity to structural changes over time (Fig. 4c). SReD effectively mapped the spatial distribution of EB3 comet activity over time. By quantifying the repetitiveness of structures, it assigned scores to different regions, highlighting areas with high EB3 comet presence and their trajectories. The NRMSE maps further emphasised this distinction, revealing elevated values along comet paths, indicative of their dynamic nature. In contrast, the Microtubule Organizing Center (MTOC) demonstrated notably lower NRMSE, suggesting its greater stability compared to the more mobile EB3 comets (Fig. 4d). The time interval used in the analysis captures the slower dynamics of EB3 comets in this context. While individual comet tracking is not the primary focus of this method, the approach effectively reveals the spatiotemporal stability of structures, where instability often results from displacement, visually manifesting as comet trajectories. To further validate our approach, we performed the analysis with increased temporal resolution. We compared SReD's results with conventional time projections of the input data, revealing advantages of our method. Unlike time projections, which typically integrate local intensities across time, SReD calculates local correlations of images across time, providing relative repetition scores that indicate how much the texture at each location changes relative to all other textures. This approach offers two significant benefits: (i) it provides a more nuanced measure of structural stability over time, and (ii) it is less

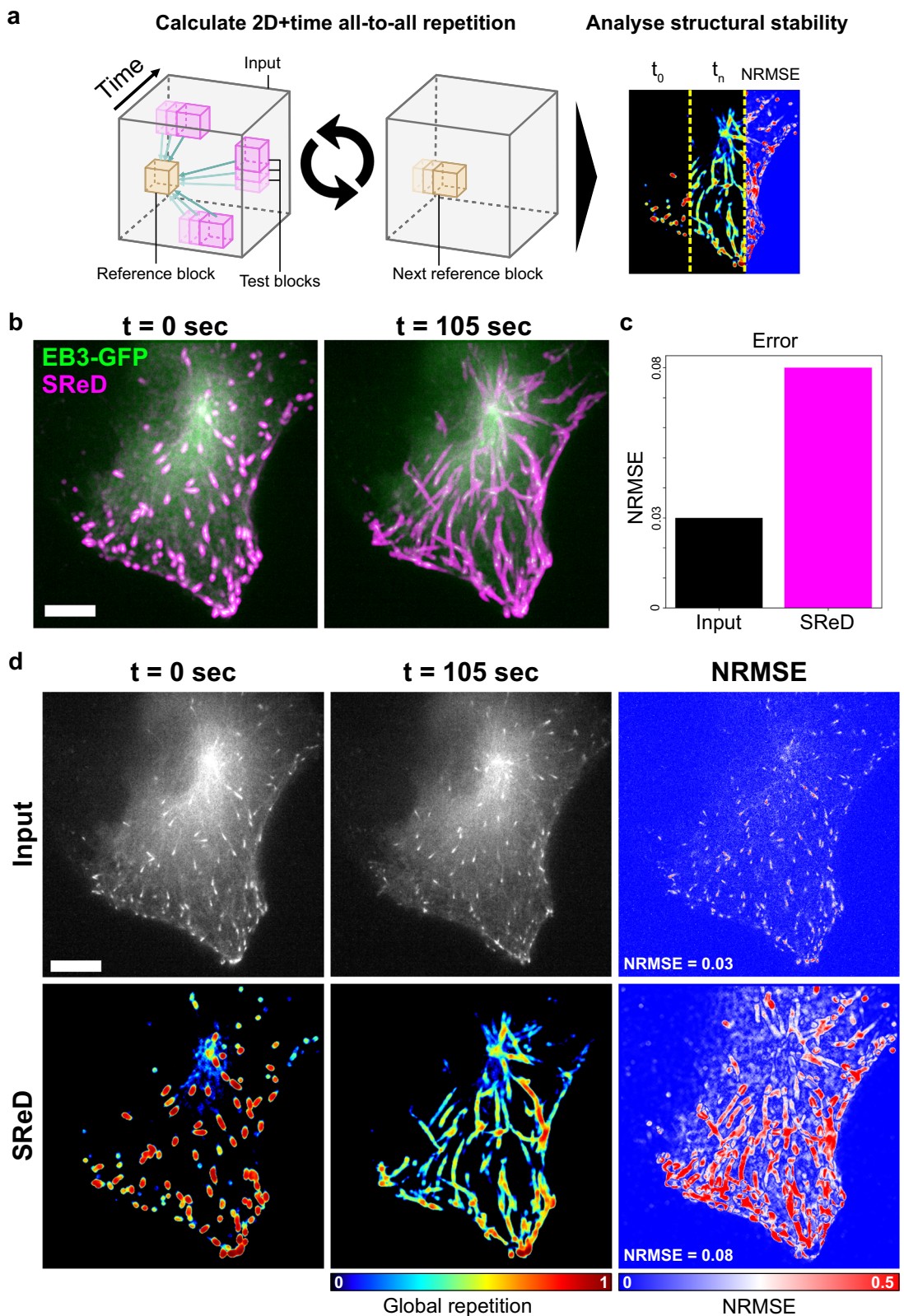

susceptible to noise and intensity inconsistencies across time points (Supplementary Note 7; Supplementary Fig. 15). In this type of combined spatial and temporal analysis, instead of producing a time series, SReD's output is a single map that shows the local stability of the timelapse over a specific time interval. This representation offers a comprehensive view of structural dynamics that is not easily achieved using traditional methods such as kymographs. While kymographs are

useful for tracking individual structures over time, SReD provides a broader perspective on the overall stability and dynamics of sub-cellular structures across the entire field of view.

## Discussion

In neuronal axons, SReD enabled automated, unbiased mapping of the membrane-associated periodic scaffold (MPS), revealing nuanced

**Fig. 4 | Assessment of the microtubule network's stability along time using SReD. a** Analysis pipeline schematic. Global repetition analysis used time as the third dimension on a timelapse sequence of RPE1 cells expressing EB3-GFP over 105 s (35 frames) ('absolute difference of standard deviations' metric). The first frame served as the control. Normalised Root Mean Squared Error (NRMSE) quantified structural differences between time points. **b** Overlay of input images (green) and temporal global repetition maps (magenta) at $t = 0$ (left) and $t = 105$ s (right). The first frame's repetition map highlights EB3 comets, while the entire time-lapse map shows comet trajectories and repetition over time. **c** Bar graph of average NRMSE between input images and repetition maps. A higher error in the repetition maps (0.08) vs. control images (0.03) indicates greater sensitivity to structural changes. Source data are provided as a Source Data file. **d** NRMSE maps of input images (top row) and repetition maps (bottom row), showing structural stability over time. High NRMSE values (warmer colours) in EB3 trajectories indicate lower stability, while lower values (cooler colours) in the Microtubule Organising Centre (MTOC) indicate higher stability. Scale bars: 10 μm.

changes in pattern frequency and prominence following pharmacological perturbation. While previous studies by Vassilopoulos et al.[19] reported a 40% reduction in overall MPS prominence after treatment with swinholide A, SReD's analysis provides a more detailed characterisation of the phenotype. By first distinguishing between regions with and without periodic patterns, and then analysing only the pattern-present areas, SReD detected a 12% reduction in pattern prominence and a 32% reduction in pattern frequency. This refined analysis not only corroborates the previously reported overall effect but also decomposes it into two distinct components, offering deeper insights into the nature of the structural changes. This analysis showcased how SReD offers a distinct approach for studying periodic structures compared to other methods, such as that described by Barabas et al.[24]. For example, while both methods rely on prior estimation of axon orientations to facilitate ring pattern detection, they differ in their implementation. Barabas et al.[24]. employ the Hough transform algorithm[25] to determine the orientation of axon edges, while SReD employs skeletonization and compares skeletons against reference blocks containing lines at different angles. In defining periodic patterns, Barabas et al.[24]. analytically model the periodic ring structure with an equation that characterizes its features. This equation is then used alongside Pearson correlation to identify regions containing the pattern. In contrast, SReD defines periodicity using image-based reference patterns, enabling comparisons based on actual structural representations rather than mathematical abstractions. This approach potentially makes SReD more adaptable for detecting irregular or complex motifs that are difficult to describe analytically. Thus, while Barabas et al.[24]. analytical approach may be advantageous for scenarios requiring precise mathematical descriptions of periodicity, SReD's image-based framework offers greater flexibility for extending its application to diverse structural patterns. These differences underscore the complementary nature of the two methods, with each suited to specific research goals and datasets.

Our results demonstrate SReD's versatility and analytical power across diverse biological contexts. For HIV Gag assembly, SReD achieved sensitive detection without relying on structural priors, significantly outperforming direct analysis of input images. This capability is particularly valuable in studying dynamic cellular processes like viral assembly, where the ability to accurately detect and characterise structures amidst variable backgrounds is crucial. In live-cell imaging of microtubule dynamics, SReD's multidimensional capabilities allowed for quantitative assessment of structural stability across space and time. This analysis provided insights into the differential dynamics of EB3 comets and the microtubule organising centre, demonstrating SReD's potential for studying complex, time-dependent cellular processes.

A key advantage of SReD is its ability to detect and characterise structures without the need for extensive labelled training data or single-molecule localisation input. This feature is particularly useful for exploratory analysis of complex biological systems where the underlying structural patterns may not be fully known a priori. The framework's flexibility in accommodating different reference blocks, from simulated idealised structures to patches extracted from the image, enhances its utility across diverse experimental scenarios. Importantly, this strength also introduces a dependence on pre-processing steps

and the choice of the reference block, which can significantly influence the output. Careful optimization of these factors is essential to ensure reliable and interpretable results. Another crucial feature is SReD's robustness to noise and pattern deformations, as demonstrated in our sensitivity analyses. This resilience enables reliable structure detection and quantification even in challenging imaging conditions, expanding the range of biological questions that can be addressed through quantitative image analysis. The algorithm's multiscale mapping capabilities provide a unique perspective on hierarchical structural organisation, as exemplified by our analysis of nuclear pore complexes at different spatial scales. While SReD offers significant advantages, it is important to acknowledge its limitations. The algorithm's performance can be influenced by the choice of reference blocks, requiring their optimisation. Additionally, while SReD reduces the need for manual region selection, some level of results curation may still be necessary, particularly in complex or heterogeneous samples. Furthermore, SReD utilises correlation metrics, each offering distinct advantages and limitations. For example, metrics such as the Pearson correlation coefficient and the Structural Similarity Index Measure (SSIM) produce satisfactory results but exhibit sensitivity to rotational variations. Conversely, metrics that demonstrate rotational invariance, such as the absolute difference of standard deviations, do not possess comparable sensitivity for detecting structural nuances. Finally, the algorithm's computational complexity warrants attention. Consider a 2D image with dimensions $n_1 \times n_2$ pixels and a block of size $k_1 \times k_2$ pixels. Each pairwise comparison between the block and an image region requires $O(k_1 k_2)$ operations. The total number of such overlapping image regions is $(n_1 - k_1 + 1)(n_2 - k_2 + 1)$. Consequently, the '1-to-all' scheme (block repetition) entails a computational complexity of $O((n_1 - k_1 + 1)(n_2 - k_2 + 1)k_1 k_2)$. When the image dimensions significantly exceed the block size ($n_1 \gg k_1$ and $n_2 \gg k_2$), this simplifies to $O(n_1 n_2 k_1 k_2)$. In the 'all-to-all' scheme (global repetition), the computational complexity scales quadratically with the image size and linearly with the block size, resulting in $O(n_1^2 n_2^2 k_1 k_2)$. SReD mitigates this computational burden by harnessing GPU acceleration and pre-calculating background regions that do not warrant analysis.

Future developments of SReD could focus on further automating the reference block selection process, potentially incorporating machine learning approaches to optimise block parameters based on image characteristics. Integration with other computational tools, such as deep learning-based segmentation algorithms, could also enhance SReD's capabilities for more comprehensive structural analysis pipelines. To address the trade-off between rotational sensitivity and structural detail detection, future work could enhance the SReD pipeline by incorporating rotation-aware analysis. This could involve systematically rotating reference blocks when employing rotation-sensitive metrics, enabling SReD to retain the high structural sensitivity of metrics like the Pearson's correlation or SSIM while mitigating their rotational limitations. Although this approach is explored in the present study, further optimization could refine the pipeline by automating processes such as image transformations. This automation would not only improve efficiency but also reduce user intervention and potential variability, broadening SReD's applicability across diverse biological contexts with varying structural orientations.

## Methods

### Optimisation of block parameters for ring pattern detection

A collection of 248 testing blocks incorporating various combinations of inter-ring spacing and ring height was generated. This process was automated using a custom ImageJ[8] macro. To create input images, five representative segments from the distal axons within each dataset were randomly selected, comprised of six neurons per treatment. These regions were then rotated to align with the horizontal axis to guarantee consistency across subsequent calculations. Then, SReD was used to generate repetition maps for every test block, and their autocorrelation functions were calculated. The relative amplitude of the autocorrelations' first harmonic was used to assess how effectively each block captured the underlying periodic pattern. The set of block parameter values that maximised the first harmonics' relative amplitude was systematically identified. The optimised set of parameter values served as a reliable representation of the periodic pattern within the dataset. The optimisation was performed separately for each dataset analysed in this study.

### Detection of reference and virus-like particles using Global Repetition

An image volume containing 3D simulated reference particles was generated using a custom Python script. The reference particles were added to the input volume by addition. Global Repetition maps were calculated using a block size of $5 \times 5 \times 5$ pixels and a relevance constant of 0. Then, the repetition maps were non-linearly mapped using a power transformation with an exponent of 10000. 3D maxima were calculated using the ImageJ[8] '3D Maxima Finder' plugin, which computes local maxima in 3D space using a flooding-based approach, with an XY and Z radius of 5 pixels and a minimum threshold of 0.1. The comparison of coordinates between the 3D maxima calculated and the reference particles was performed using a custom Python script.

### Cell culture

Jurkat cells were cultured in RPMI 1640 (Gibco) supplemented with 10% fetal bovine serum (FBS), 2 mM L-glutamine and 50 μg/mL gentamycin. HEK293T and RPE1-EB3-GFP cells were cultured in DMEM supplemented with 10% fetal bovine serum (FBS), 2 mM L-glutamine and 50 μg/mL gentamycin. Cell lines were cultured at 37 °C and 5% $CO_2$.

### DNA plasmids and cell lines

The RPE1-EB3-GFP cell line was kindly provided by Dr. Mónica Bettencourt-Dias. A plasmid expressing HIV Gag with an internal EGFP tag was generated using the NEB HiFi Assembly Kit (New England Biolabs). A lentiviral backbone containing a tetracycline-inducible promoter and a gene encoding rtTA was prepared by digesting the pCW57.1 plasmid (Addgene #41393) with 5 μg/mL restriction enzymes BamHI and NheI (New England Biolabs) according to the manufacturer's instructions for 1 h at 37 °C. The digestion product was separated using 1% agarose gel electrophoresis (AGE) and the 7.6˜ kb band was purified using the GFX PCR & Gel Band Purification Kit (Sigma-Aldrich) according to the manufacturer's instructions. Then, three DNA fragments were generated by polymerase chain-reaction (PCR) using Q5 High-Fidelity DNA Polymerase (New England Biolabs). The first fragment (445 bp), encoding the HIV-1 Matrix protein followed by an HIV-1 protease cleavage site (MA-PCS), was generated using Optigag-mNeonGreen-IN[26] as a template and primers *5′-tcagatcgcctggagagaattgggccaccatgggtgcgcga3′ (Fw)* + *5′-ccatacgcgtctggacaatggggtagtttttgactgacc-3′ (Rv)*. The second fragment (751 bp), encoding EGFP, was generated using HIV-(i)GFP ΔEnv[21] as a template and primers *5′-ccattgtccagacgcgtatggtgagcaag-3′ (Fw)* + *5′-tagttttgacttctagacttgtacagctcgtc-3′ (Rv)*. The third fragment (1.2 kb), encoding a PCS and the HIV-1 Capsid, Nucleocapsid and p6 proteins (PCS-CA-NC-p6), was generated using Optigag-mNeonGreen-IN[26] as a template and primers *5′-caagtctagaagtcaaaaactaccccattgtc-3′ (Fw)* + *5′-aaaggcgcaaccccaaccccgtcattgtgacgaggggtctgaac-3′ (Rv)*. The

three fragments were purified using DNA purification columns and their molecular size was confirmed by AGE. The HiFi Assembly reaction was performed using 50 ng of digested vector and equimolar amounts of the three fragments and incubated at 50 °C for 1 h. The reaction product was diluted 1:4 in dH20, and 2 μL of the dilution was transformed into chemically competent STABL4 bacteria (Thermo Fisher). The bacteria were plated in LB-agar supplemented with 100 μg/mL ampicillin and incubated overnight at 37 °C. Several colonies were picked and inoculated into liquid LB containing ampicillin at 100 μg/mL. The plasmid DNA from these colonies was extracted using the GenElute Plasmid Miniprep Kit (Sigma-Aldrich) and was confirmed by digestion with restriction enzyme XbaI followed by AGE (2.3 kb and 7.5 kb fragments). A positive colony was then sequenced using Sanger sequencing (Genewiz) and primers *5′-cgtcgccgtccagctcgacca3′*, *5′-ccattgtccagacgcgtatggtgagcaag-3′* and *5′-aaaggcgcaaccccaaccccgtcattgtgacgaggggtctgaac-3′*. This process yielded the lentiviral plasmid TetOn-Optigag(i)EGFP, where a human codon-optimised *gag* gene contains a PCS-flanked EGFP-encoding gene. Lentivirus packaging TetOn-Optigag-(i)EGFP were produced to transduce Jurkat cells. To do this, HEK293T cells were cultured in 6-well plates until ~80% of confluence, transfected using 300 μL/well of transfection mixture (DMEM, 3 μg of TetOn-Optigag-(i)EGFP, 1.5 μg of psPAX2 (Addgene #12260), 1.5 μg of CMV-VSV.G (NIAID) and 12 μL of linear polyethyleneimine MW-25,000 (final concentration of 5 μg/μL)(Sigma-Aldrich)) and incubated overnight for 8 h. Then, the culture medium was replaced with complete DMEM, followed by a 24-h incubation. The virus-rich supernatant was collected and filtered with 0.22 μm syringe filters. Jurkat cells (2 mL at $1 \times 10^6$ cells/mL) were inoculated with 300 μL of virus-rich supernatant and Polybrene (10 μg/mL), followed by a 3-day incubation. Antibiotic selection of transduced cells was performed by replacing the culture medium with complete RPMI containing puromycin at 2 μg/mL and incubating for 3 days, at which point an 'empty virus' control sample had no live cells remaining. The cells were incubated with doxycycline at 1 μg/mL for 24 h to induce expression and single cells were isolated using Fluorescence-assisted Cell Sorting (FACS). The EGFP-positive population was divided into three subsets according to their relative signal intensity ('Low', 'Medium' and 'High') and single cells were plated into 96-well plates. The cultures were expanded for 15 days, and the resulting cell lines were validated using fluorescence microscopy and Western blotting. A clonal line of the 'Medium' subset was used for this study.

### Sample preparation and acquisition of microscopy data

**HILO imaging of HIV virus-like particle assembly in activated Jurkat cells.** Activation surfaces were prepared based on the protocol in ref. 27 To do this, Lab-Tek 8-well chambers (Thermo Fisher) were cleaned with 100% isopropanol for 10 min and followed by three washing steps with $dH_2O$. Then, 200 μL of a mouse anti-CD3 antibody diluted in PBS at a final concentration of 1 μg/mL was added to the wells and incubated overnight at 4 °C. The wells were carefully washed twice with PBS to remove unbound antibodies. Jurkat cells expressing TetOn-Optigag-(i)EGFP were incubated with 1 μM of doxycycline (Sigma-Aldrich) for 24 h. Then, 50,000 cells were added to each well and allowed to adhere and stabilise for 1 h. Imaging was done in a Nanoimager (ONI) using the 488 nm laser at 10% and channel 0 (two-band dichroic: 498–551 nm and 576–620 nm). The HILO angle was optimised manually, and images were acquired at 100 ms exposure. Pixel size: 117 nm. The anti-CD3 antibody was produced at the Flow Cytometry & Antibodies Unit of Instituto Gulbenkian de Ciência, Oeiras, Portugal.

**3D imaging of HIV virus-like particle assembly in activated Jurkat cells.** Jurkat cells expressing TetOn-Optigag-(i)EGFP were centrifuged at $200 \times g$ for 5 min and resuspended in complete RPMI containing 0.5 μM of doxycycline to induce Gag expression. Glass coverslips (1.5 mm thick, round, 18 mm diameter) were

washed with isopropanol for 10 min followed by three washes with $dH_2O$. The coverslips were coated with Poly-L-Lysine (PLL, Sigma Aldrich) at 0.1% and incubated for 15 min at room temperature, followed by three washing steps with $dH_2O$. The PLL-coated coverslips were dried, mounted in an Attofluor chamber (Thermo Fisher) and fixed on the microscope's stage. The microscope's enclosure (Okolabs) was heated at 37 °C and a manual gas mixer (Okolabs) was used to supply 5% $CO_2$. The cells were seeded in the pre-treated coverslips and allowed to settle in the microscope enclosure for 30 min. Imaging was performed on an inverted microscope ECLIPSE Ti2-E (Nikon Instruments) equipped with a Fusion BT (Hamamatsu Photonics K.K., C14440-20UP) and a Plan Apo λ 100× (NA 1.45) Oil objective. The sample was illuminated with LED light at 515 nm (CoolLED pe800) and acquisition was done at 75 ms exposure with an active Nikon Perfect Focus system and the NIS-Elements AR 5.30.05 software (Nikon Instruments). Volumes were captured by acquiring frames at different depths (z-step size: 0.5 μm). Image deconvolution was performed using a custom Python script based on the Richardson-Lucy method[28,29] as described in refs. [30], [31].

**Imaging EB3-GFP comets in RPE1 cells.** RPE1-EB3-GFP cells (50000 per well) were seeded into Lab-Tek 8-well glass chambers (Thermo Fisher) and allowed to adhere for 24 h. Widefield imaging was performed in a Nanoimager (ONI) using the 488 nm laser at 10% and channel 0 (two-band dichroic: 498–551 nm and 576–620 nm). Images were acquired at 75 ms exposure for 2 min. Pixel size: 117 nm.

**Assessment of the microtubule network's stability along time using SReD.** Subsets of the original time lapse were created by keeping images belonging to the time frames of interest. Global repetition maps were generated from the temporal subsets using an XY block size of 7 × 7 pixels, a Z block size equal to the number of images in each subset, and a relevance constant of 0. The repetition maps were non-linearly mapped using a power transformation with an exponent of 1000. NRMSE maps were calculated using the 'scikit-image' library (v0.22.0, accessible at https://scikit-image.org/docs/stable/api/skimage.metrics.html).

### Statistics and reproducibility
No statistical method was used to predetermine sample size. No data were excluded from the analyses. The experiments were not randomized. The Investigators were not blinded to allocation during experiments and outcome assessment.

### Reporting summary
Further information on research design is available in the Nature Portfolio Reporting Summary linked to this article.

## Data availability
The data obtained in this study is available at https://doi.org/10.5281/zenodo.13764726 and https://doi.org/10.6019/S-BIAD1620 under CC BY 4.0 license. The STORM data containing cells with labelled microtubules is available at https://doi.org/10.5281/zenodo.5534351[12] The widefield microscopy data containing DAPI-stained nuclei is available at https://doi.org/10.5281/zenodo.3232478[13] The STORM data containing nuclear pores with labelled gp210 is available at https://www.embl.de/download/ries/excitation_intensities/Nup96-BG-AF647_250kWcm2_57_2.zip[14]. Source data are provided with this paper.

## Code availability
The SReD algorithm, along with all custom scripts used in this manuscript are available at http://github.com/HenriquesLab/SReD (release v1.0). All source code is under an MIT License.

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

## Acknowledgements

A.M. thanks Simão Coelho and Estibaliz Gómez-de-Mariscal for their feedback during the formulation of the algorithm and its demonstration in biological studies. A.M. and R.H. acknowledge the support of the Gulbenkian Foundation (Fundação Calouste Gulbenkian), the European Research Council (ERC) under the European Union's Horizon 2020 research and innovation programme (grant agreement No. 101001332) (to R.H.) and funding from the European Union through the Horizon Europe program (AI4LIFE project with grant agreement 101057970-AI4LIFE and RTSuperES project with grant agreement 101099654-RTSuperES to R.H.). Funded by the European Union. Views and opinions expressed are, however, those of the authors only and do not necessarily reflect those of the European Union. Neither the European Union nor the granting authority can be held responsible for them. This work was also supported by a European Molecular Biology Organization (EMBO) installation grant (EMBO-2020-IG-4734 to R.H.), a Chan Zuckerberg Initiative Visual Proteomics Grant (vpi-0000000044 with https://doi.org/10.37921/743590vtudfp to R.H.) and a Chan Zuckerberg Initiative Essential Open Source Software for Science (EOSS6-0000000260). This study was also supported by the Research Council of Finland (338537 to G.J.), the Sigrid Juselius Foundation (to G.J.), the Cancer Society of Finland (Syöpäjärjestöt; to G.J.), the Solutions for Health strategic funding to Åbo Akademi University (to G.J.), the InFLAMES Flagship Programme of the Academy of Finland (decision numbers: 337530, 337531, 357910 and 357911). This research was also supported by the National Institutes of Health (NIH) with grants K22AI140963, K61DA058348 and subcontract R01AI50998 (to J.I.M). C.L. acknowledges funding from the Agence National de la Recherche (ANR-20-CE13-0024 'ASHA', ANR-21-CE42-0015-01 '5D-SURE'). C.L. acknowledges the INP NCIS imaging facility and Nikon Center of Excellence for Neuro-NanoImaging for service and expertise, with funding from Excellence Initiative of Aix-Marseille University, A*MIDEX, a French 'Investissements d'Avenir' program (AMX-19-IET-002) through the Marseille Imaging and NeuroMarseille Institute.

## Author contributions

A.M. and R.H. conceived the study in its initial form; A.M. and R.H. developed the SReD framework with code contributions from B.M.S.; B.M.S., G.J., J.I.M and C.L. provided samples, data, critical feedback, testing and guidance; A.M. performed experiments and analysis; G.J., J.I.M., C.L. and R.H. acquired funding; C.L., R.H. supervised the work; A.M and R.H. wrote the manuscript with input from all authors.

## Competing interests

The authors declare no competing interests.
