## [Transparent Peer Review file · Nature Communications]

Structural Repetition Detector for multi-scale quantitative mapping of molecular complexes through microscopy

Corresponding Author: Professor Ricardo Henriques

Version 0:

Reviewer comments:

Reviewer #1

(Remarks to the Author)

In the paper entitled: "Structural Repetition Detector: multi-scale quantitative mapping of molecular complexes through microscopy" Mendes et al. propose an original image analysis method for the detection of recurrent patterns in microscopy datasets. The proposed method is an unsupervised computational framework that can be either fed with sample images (both simulated or experimental) or run freely on the dataset to find emerging patterns. Repeated features can be found at different scales depending on the reference block size and features. The authors validate the analysis method on diverse kinds of datasets with both ad hoc designed reference blocks or without any user-provided reference. Notably, the authors demonstrate that the Global Repetition Maps produced by the analysis can be very powerful in the structural analysis of the images even compared to original data, and give different examples to support this.

The proposed method gives many valuable advantages in terms of the versatility, the improved output given by the Global Repetition Maps analysis, and its robustness against SNR degradation and image deformations. On the other side, as the authors already highlight in the discussion, it poses some operating issues in terms of time of operation that should be carefully taken into account and handled when possible, and the strong dependence on the Reference Block features set by the user.

In my opinion, the possibility to apply a general, versatile and reliable method to find recurrent patterns in large data sets is very valuable to perform quantitative analysis on microscopy images. For this reason, I think this article could be of great interest for Nature Communications readers community. Nevertheless, I strongly suggest some revisions prior to publication that I list here below.

Major revisions

General comments:

The reader is scattered from the main text to the supplementary information and to the methods section too many times, making the reading of the article very difficult. The authors may try to put all the comprehensive information on the algorithms in the Supplementary Information, while leaving only the experimental details in the Methods. If not, they should find another way to avoid excessive fragmentation of the information. Also, the image captions should be revised to be more self-consistent, hence much more informative. Pay attention to acronyms, which are sometimes defined first in the figure captions or in the Supplementary Info, while they come out of the blue in the main text. Keep in mind that, despite its complexity, the reader should be able to understand the proposed method from the main body of the manuscript.

Specific comments:

1) The method proposed by the authors is very powerful, mainly due to the incredible versatility, but this feature in my opinion could represent also its weakness, being the output results very dependent on i) the pre-processing of data and, in particular, on ii) the choice of the reference block made by the user. This is particularly evident in the example of the repetition map in neurons and should be expanded in the conclusions of the article, together with the advantages of the method which are already fully discussed. Also, in the application to microtubule dynamics the authors could discuss how results would be affected by the choice of reference blocks of different durations.

2) In the Results, in the 3D case example, the authors should specify what they mean for 3D images. In the methods, the authors describe the data set as composed by images taken on a widefield epifluorescence microscope, at different depths, with 0,5 microns distance in between consecutive planes. This means that technically speaking the single frames acquired are bidimensional. If this is the case, the authors should clearly specify this in the main body, to clarify that the single frames are not three-dimensional, and the z coordinate is given by the position in the stack of images. In this regard, they should also clarify what the function "3D local maximum" is doing, and specify whether the z position is and output of this function or not. Currently, the z position is reported in Figure 3 but it is not explained in the body, nor referenced to the methods section. In general, this section is quite confusing, the strategy of adding artificially added reference particles should be better explained, also in the caption of Fig. 3.

#Minor revisions:

- 1) In Figure 1 fonts are too small.
- 2) In the results, section on nuclear envelope analysis, authors state: "We further illustrate SReD's versatility by detecting nuclear envelopes in DAPI- stained cells (11) using empirical reference blocks extracted directly from the input image, distinguishing different morphological states potentially related to cell division or stress (Fig. 1b; Fig. S3)." This is not clear to me, please explain better. A reference to Figure S3d could be more appropriate, since there the differences in the nuclear shapes are more evident and quantified.
- 3) In the Methods, section "Relevance mask", the authors state "This is obtained by sampling the variance of the input image using non-overlapping blocks of the same size as those used for the repetition analysis, and then calculating the average of percentile 0.03.". The authors should discuss the reason for the number 0.03.
- 4) In the Supplementary Information, Note S1, point 2. Generation of the Relevance Mask, the authors should expand it to be more self-consistent, or at least refence to the Methods. Please define the constant "C" that is found unreferenced in the caption of figure S1.
- 5) In Note S1, when introducing the SRS the authors should specify that the range is form 0 to 1, of course this is evident from the pictures but should be defined.
- 6) In Note S2, the authors should put a reference for the "Cosine similarity " function or explain briefly how it works.
- 7) In Note S3, section "Detection of ring patterns in large fields-of-view", point 3. Generation of angular weight maps, the authors should better explain the weighting procedure, this is not clear at all.
- 8) In Note S4, section "Analysis of SReD's robustness against non-specific structures", the detection counts out of the global repetition maps does not show a linear trend, while this is the case of the input image counts. Can the authors comment on this difference?

(Remarks on code availability)

Reviewer #2

(Remarks to the Author)

(Remarks on code availability)

Reviewer #3

(Remarks to the Author)

Dear Authors,

General remarks

The authors present a Fiji plugin for the (semi-)automated detection of repetitive structures in images. We think this could become an important tool, especially for imaging data aiming at resolving structural details by averaging multiple occurrences of the same entity in one image dataset.

This is interesting for super-resolution fluorescence microscopy but also for cryo-ET data. Regarding the latter we would like to suggest the authors to explore their tool also on a cryo-ET image, e.g. identifying the locations of ribosomes or other protein complexes.

Regarding exploring an image dataset: As will be mentioned below we did not yet manage to actually run the software, but at a first glance it was not obvious to us how to eventually run it across many images of the same sample. Would that be possible?

Comments about the manuscript

In general we found that the main text did not contain enough detail and it was not possible to understand what the method is actually doing. We would recommend to move quite a lot of the methods section into the main text and then have less examples in main text. In fact, some of the examples of running the algorithm on different dataset could be in the supplemental material.

To give a few examples:

“The analysis proceeds using reference blocks, either simulated or empirically sampled from the image.” It is not explain what “empirically sampled” means.

“These blocks are matched against the input using correlation metrics to generate repetition maps, as detailed in the Methods section.” We think it is better to explain this here in the Results section and not only in the Methods.

“We measured an average spacing of 178 nm under control conditions.” How was this measured?

“Nonlinear mapping effectively distinguishes real patterns from imprinted ones.” The explanation of “nonlinear mapping” is missing.

Other comments

“Localisation of round viral structures via local extrema calculation revealed that the repetition map provided a superior platform for extrema detection compared to direct analysis of the raw images.”

Please add a reference to a corresponding Figure.

Clarify interpretation of ref 7,8: As far as we know these methods don't need single molecule localization data, they work with any set of points.

Data availability

In general, for the image data we recommend publication in the BioImageArchive, because it is searchable and will allow to provide a minimal amount of imaging relevant metadata.

For data that is reused from existing publications, we think it is not sufficient to link to the corresponding publication, but the link must directly go to the image data. Otherwise it is not practical for the reader to actually find the data in a realistic amount of time.

Method section

We found that the method in general looks very much like a standard template matching approach. Please discuss/compare with other related methods (e.g. cross-correlation based template matching)? For example, how does the proposed method differ from existing implementations such as https://scikit-image.org/docs/stable/auto_examples/features_detection/plot_template.html ?

How is noise defined for the GAT step?

“and then calculating the average of percentile 0.03.” Please rephrase, what do you mean by the average of a percentile?

“This produces a binary mask outlining areas with sufficient structural content.” Add link to a Figure.

“"1-to-all" or "all-to-all" “: Consider renaming to “Manual Reference” and “Automatic Reference” ?

“modified cosine similarity”: Please explain the maths.

“To minimise blocking artefacts, the square blocks are transformed into their inbound elliptical counter-parts”?: Please explain in more detail.

“In the global repetition mode, SR ...”: Is “global repetition mode” the same as “all-to-all”?

“Non-linear mapping”: We wondered whether the non-linear mapping is mathematically similar to the above W weight function? If so, could this be condensed into one step in the algorithm?

“This process was automated using a custom ImageJ macro.” Please add a link to the corresponding code repository.

“An image volume containing 3D simulated reference particles was generated using a custom Python script.” Please add a link to the corresponding code repository.

“The repetition maps were non-linearly mapped using a power transformation with an exponent of 1000.

NRMSE maps were calculated using the "scikit-image" library (v0.22.0)." Please add a link to the corresponding code repository.

How are borders defined in computing N in equation 5 ("N is the size of the input image excluding borders")?

Please discuss whether rotated particles would also be detected? A simple cross-correlation would not do this but maybe the authors framework takes rotations into account?

Please also discuss robustness to: Structures of different sizes? Partial occlusion? Deformations?

Software

We did not yet manage to run the software but we are very happy to test a revised version.

For issues installing the software, see <https://github.com/HenriquesLab/SReD/issues/3>

There is a lack of feedback during long computations, e.g. when doing variance stabilisation, nothing visible happens though eventually a new image window pops up. Please add progress reports (ImageJ has a progress bar that could be used).

As mentioned above, is there support for the analysis of multiple images of the same sample?

Best wishes,

Christian Tischer and J-K Heriche

(Remarks on code availability)

We tried running the software but ran into some issues (see above).

Reviewing the actual code is something that we could offer for a follow-up revision of the article.

Reviewer #4

(Remarks to the Author)

Mendes et al described SReD, an unsupervised computational framework that enables structure detection in microscopy images by finding local structural repetition. In addition to the principle of structure detection in the SReD, four small examples in Fig. 1 and three examples in Fig. 2-4 were used to demonstrate the abilities of SReD to detect repetitive structures using data quantitated by this program. In particular, in Fig. 2 and its supplemental figures, the program succeeded in quantifying various parameters that were consistent with the original article and newly extracted in this study. This series of analyses indicates the utility of SReD. In contrast, the other two examples are less clear about what information users can extract from the data. In this context, I suggest that the authors consider the following points.

Major points

- In Fig. 3, the authors demonstrate that the program can detect the positions, intensities, and numbers of HIV-1 Gag assemblies in 3D. However, since the virus particles are estimated to be 100-150 nm in diameter, which is below the diffraction limit, recent studies on the virus assemblies have been performed using super-resolution microscopy. therefore, readers would be interested in the ability of SReD to analyze Gag assemblies in super-resolution microscope images, such as the size estimation of assemblies, etc. It would be great to analyse published super resolution microscopy data or STORM data obtained in the authors' system.
- Alternatively, it would be interesting and helpful for future users to show similar data using Gag-(i)EGFP with multimerization deficient mutations (e.g. W185A W186A), as the authors demonstrated EGFP intensity of Gag-(i)EGFP as a function of SRS.
- In Fig. 4, the authors showed that the EB3 trajectory indicated lower stability, while MTOC indicated higher stability. This conclusion is intuitively known and not surprising when comparing moving spots and static structures, although I recognize SReD's utility in this example, as shown in Fig. S13. I wondered if the program could calculate a EB3 velocity using a combination of the SReD's functions.

Minor points

- The Figure legends in Fig. 3 and 4 do not seem to correspond to the figures.

(Remarks on code availability)

Version 1:

Reviewer comments:

Reviewer #1

(Remarks to the Author)

Dear Editor, dear Corresponding Author,
after thoroughly reviewing the revised manuscript, I believe that the authors have substantially addressed the concerns I raised in my previous feedback. In its current form, the manuscript effectively conveys the potential of the proposed method and its significance.

Given these improvements, I recommend the manuscript for publication in Nature Communications.

Thank you for the opportunity to review this work.

Best regards,

Lucia Gardini

(Remarks on code availability)

I didn't try the code myself.

Reviewer #2

(Remarks to the Author)

(Remarks on code availability)

Reviewer #3

(Remarks to the Author)

We would like to thank the authors for taking the time to address many of our comments!

While the manuscript improved in terms of detailed points we still think that the overall structure should be changed.

In this regard we agree with Reviewer 1's comment about the first version of the manuscript: "The reader is scattered from the main text to the supplementary information and to the methods section too many times, making the reading of the article very difficult. The authors may try to put all the comprehensive information on the algorithms in the Supplementary Information, while leaving only the experimental details in the Methods. If not, they should find another way to avoid excessive fragmentation of the information."

Relatedly, we wrote: "In general we found that the main text did not contain enough detail and it was not possible to understand what the method is actually doing. We would recommend to move quite a lot of the methods section into the main text and then have less examples in main text. In fact, some of the examples of running the algorithm on different dataset could be in the supplemental material." (We could not find this section in the rebuttal letter and thus it may have been lost at some point and not accessible to the authors. We are sorry if this was our mistake.)

In our view, this article is about a new analysis method and thus we think this method should be properly described in the main text. There can also be application examples in the main text, but (some of them) could also be in the supplemental material. We think, for a methods paper, there is too little technical information in the main text. We thus still feel that it would be better to rewrite the article such that more technical information is presented in the main text and some of the examples are moved into the supplemental material.

Specifically:

Line 97: "using correlation metrics" does, in our view not contain enough information. As mentioned in our first review, we feel that it should be explained early within the main text how these correlations are computed and what the difference is to simple standard cross-correlation based approaches, which are available in standard libraries, and also as ImageJ plugins such as <https://bmcbioinformatics.biomedcentral.com/articles/10.1186/s12859-020-3363-7>

Specifically, there are different modes that are more or less rotation invariant. It is critical to explain this and at least some of the mathematical foundation to the reader. At line 340 the different correlation metrics are discussed, but since they were not introduced in the main text, one cannot really understand the discussion.

Also when demonstrating the method for different example data, it should be discussed in the text which correlation metrics have been used and why, such that users of the method can make informed decisions.

In addition to the correlation metric the other main parameter is the size of the "correlation box"; also here it would be important to add guidelines into the article as to how big this should be chosen and also what happens if one chooses the wrong size.

Relatedly, the "theoretical foundation" section actually starts with implementation details (the fact that this is available as Fiji

plugin is, in our view, not a theoretical foundation, the same ideas could be implemented in other platforms). In addition, please explain what a relevance mask is and how it is computed. At the minimum, put a summary of note S1 in the main text and the content of note S1 into the method section. As this is a methods paper, the reader should not need to dig in supplementary docs to get all details.

Test of the plugin

We now managed to run the plugin!

Noise variance stabilisation

This takes several seconds and indication of progress or whether anything is actually being done is important. We thank the authors of implementing progress in the Fiji progress bar, however, we initially did not even realise this, because the progress bar disappears as soon as the mouse is moved. We recommend also indicating progress in the IJ log window. See here how to use the `\\Update` functionality to achieve this: <https://wsr.imagej.net/macros/LogWindowTricks.txt>

The wiki appears to suggest that some (histogram? line profile?) plot should appear along the image but we only see the new image and no plot. Maybe remove the plot from the wiki if it is not critical? In addition, the example in the wiki doesn't show any difference obvious between before and after.

Relevance mask

We found this to be rather slow (even though it was using the GPU): Is it supposed to take >30s to compute for a 2000x2000 image?

Choosing the box size and ROIs

The requirement for odd numbers is cumbersome, especially when drawing a ROI to select the reference mask. We recommend to automatically select the closest odd number and display a warning about it in the log window. As it is, when one makes a mistake, one needs to go back, find the plugin in the menu and start again.

(Remarks on code availability)

Reviewer #4

(Remarks to the Author)

The utility of SReD in detecting sub-diffraction size particles, such as HIV-1 Gag assembly and virus, is well demonstrated by these new supplementary figures with different image resolutions. In new Figs. S6 (b and c) and S7a, side-by-side presentations of treated input images and the block repetition maps, would enhance clarity about how the analysis works. Figure R1 shows that SReD can output sufficient repetition maps to allow downstream analysis of the EB3 trajectory in the tracking program TrackMate, demonstrating another application of SReD data. The authors may consider including this figure in the supplementary materials and explanation in the text and the Methods section (or Supplementary Note). Furthermore, if SReD offers advantages over directly using other tracking programs, incorporating these benefits would be valuable for potential users.

(Remarks on code availability)

Dear Editorial Team and Reviewers,

We are deeply grateful for your comprehensive review of our manuscript on the Structural Repetition Detector (SReD). The review process has markedly enhanced our work's scientific rigour and clarity.

In response to the points raised, we have implemented several major revisions to improve the manuscript:

- New Supplementary Notes and Figures have been added, showcasing SReD's capability to detect and classify viral particles with sub-diffraction limit resolution. The first comprises the identification of HIV-1 particles and mature virions in electron microscopy data, a comparison with a deep learning-based method, and a video tutorial illustrating the analysis process using ImageJ/Fiji and the SReD plugin. The second comprises the detection of viral assembly platforms in STORM data, and a comparison between control cells and cells treated with an actin-debranching drug that increases the number of budding virions.
- The main text has been revised to enhance self-sustainability, reducing the need for readers to consult supplementary materials. Figure captions have been expanded to offer more comprehensive and consistent information. We have also included more detailed explanations of our methodology, including a new Results section dedicated to the method's core functionality, and a clearer explanation regarding the use of simulated reference particles for accuracy estimation. Additionally, the discussion section has been broadened to address the versatility and potential limitations of our method more thoroughly. It was also divided in subsections to improve clarity.
- We have clarified the structure of 3D data and the acquisition method for 3D images.
- We have committed to publishing our image data in the BioImageArchive to improve searchability and metadata provision. Direct links to datasets reused from existing publications have been added.
- A Supplementary Table was added. Here, we made a comparison of SReD's functionalities against other methods in the same domain, such as template matching.

The revisions enhance our manuscript's accessibility and comprehension. We believe these changes address the reviewers' concerns and improve the quality of our work. We appreciate the reviewers' time and effort in evaluating our manuscript. The following sections detail our responses to each comment.

Reviewer 1:

- Reviewer comment: The reader is scattered from the main text to the supplementary information and to the methods section too many times, making the reading of the article very difficult. The authors may try to put all the comprehensive information on the algorithms in the Supplementary Information, while leaving only the experimental details in the Methods. If not, they should find another way to avoid excessive fragmentation of the information.

Our Reply: We acknowledge the reviewer's concern regarding the fragmented structure of our manuscript. To address this issue, we have revised the main text to be more self-contained and comprehensive. The revised manuscript now presents complete explanations of key concepts, methodologies, and results within the main text, ensuring readers can fully understand the core findings without constantly referring to supplementary materials. While we maintain detailed technical information and extended analyses in the Supplementary Notes and Figures, these are now purely complementary resources for readers seeking deeper insights into specific aspects of our work.

- Reviewer comment: The image captions should be revised to be more self-consistent, hence much more informative.

Our Reply: We have revised figure captions to ensure comprehensive and self-contained descriptions of each visual element. For example, in Figure 3, we expanded the caption to specify that "Reference particles were designed as diffraction-limited particles with randomly distributed intensities across the input's dynamic range and serve as a reference with a known ground-truth to evaluate the algorithm's accuracy".

- Reviewer comment: Pay attention to acronyms, which are sometimes defined first in the figure captions or in the Supplementary Info, while they come out of the blue in the main text.

Our Reply: We have revised the manuscript to ensure proper introduction and consistency of acronyms. Each technical term is now explicitly defined upon its first appearance in the main text before being used in figure captions or supplementary materials. For instance, terms like "Structural Repetition Score (SRS)" and "Generalised Anscombe Transform (GAT)" are now properly introduced with their full terminology when first mentioned.

- Reviewer comment: Keep in mind that, despite its complexity, the reader should be able to understand the proposed method from the main body of the manuscript.

Our Reply: We enhanced the main text to ensure that readers can fully understand the method without needing to consult the Supplementary Notes and Figures. As a result, the main manuscript is now more self-contained, while the Supplementary Notes and Figures provide additional details about each section for readers who wish to explore further.

- Reviewer comment: The method proposed by the authors is very powerful, mainly due to the incredible versatility, but this feature in my opinion could represent also its weakness, being the output results very dependent on i) the pre-processing of data and, in particular, on ii) the choice of the reference block made by the user. This is particularly evident in the example of the repetition map in neurons and should be expanded in the conclusions of the article, together with the advantages of the method which are already fully discussed.

Our Reply: We agree that SReD's versatility can also represent its weakness for the reasons specified by the reviewer. In response, we have expanded our discussion to address this point more comprehensively. In the revised manuscript, we emphasise that while SReD's flexibility is a significant strength, it also necessitates careful consideration in its application. We now explicitly state that the method's output is indeed sensitive to data preprocessing and reference block selection. To mitigate this potential weakness, we have included guidance on optimising these parameters, emphasising the importance of systematic approaches to reference block design and data preprocessing. We have also added examples demonstrating how variations in these factors can affect results, providing users with a more nuanced understanding of the method's sensitivities. This expanded discussion balances the presentation of SReD's advantages with an examination of its limitations, offering readers a more comprehensive view of the method's capabilities and considerations for its effective use.

- Reviewer comment: Also, in the application to microtubule dynamics the authors could discuss how results would be affected by the choice of reference blocks of different durations.

Our Reply: We appreciate the reviewer's comment regarding the impact of reference block duration on microtubule dynamics analysis. While the main manuscript indeed focuses on a single block duration for clarity, we have expanded this analysis in Note S6 and Figure S14. These supplementary materials demonstrate how varying block durations affect the spatiotemporal analysis of microtubule dynamics. Specifically, we show that longer duration blocks capture slower, more stable dynamics (e.g., MTOC movement), while shorter duration blocks highlight rapid changes (e.g., EB3 comet trajectories). This multi-timescale approach provides a more comprehensive view of microtubule behaviour, revealing how different structural components evolve over time. We believe this additional information addresses the reviewer's concern while maintaining the focus of the main text.

- Reviewer comment: In the Results, in the 3D case example, the authors should specify what they mean for 3D images. In the methods, the authors describe the data set as composed by images taken on a widefield epifluorescence microscope, at different depths, with 0,5 microns distance in between consecutive planes. This means that technically speaking the single frames acquired are bidimensional. If this is the case, the authors should clearly specify this in the main body, to clarify that the single frames are not three-dimensional, and the z coordinate is given by the position in the stack of images.

Our Reply: We have revised the main text to provide a more precise description of our 3D imaging approach. The 3D dataset comprises a series of 2D images acquired using a widefield epifluorescence microscope at sequential focal planes, with a z-step size of 0.5 μm between consecutive images. This stack of 2D images collectively forms a 3D volume, where the z-coordinate of each slice corresponds to its position within the image stack. We have explicitly clarified this point in the main text to ensure readers understand that individual frames

are two-dimensional, and the three-dimensional nature of the data emerges from the compilation of these 2D slices at different focal depths.

- Reviewer comment: In this regard, they should also clarify what the function "3D local maximum" is doing, and specify whether the z position is and output of this function or not. Currently, the z position is reported in Figure 3 but it is not explained in the body, nor referenced to the methods section.

Our Reply: We have expanded our explanation of the "3D local maximum" function in the main text and methods section. Specifically, we clarified that this ImageJ/Fiji plugin computes local maxima in 3D space using a flooding-based approach. The function outputs the x, y, and z coordinates of detected maxima, which correspond to the centres of potential viral particles in our analysis. We have also added a reference to the methods section where readers can find more detailed information about the plugin's parameters and implementation. This addition provides a clear link between the z-position data reported in Figure 3 and the underlying analytical method.

- Reviewer comment: In general, this section is quite confusing, the strategy of adding artificially added reference particles should be better explained, also in the caption of Fig. 3.

Our Reply: We have revised our explanation of the simulated reference particles to improve clarity. In the updated manuscript, we elaborate on the rationale and methodology behind these artificial structures. The simulated particles serve as a known ground-truth, enabling a quantitative assessment of our detection method's accuracy. We designed these particles to mimic the characteristics of fluorescent viral particles observed in widefield microscopy: diffraction-limited, slightly blurred, and with intensities randomly distributed across the input image's dynamic range. This approach provides a benchmark for evaluating the performance of our detection algorithm under controlled conditions. We have incorporated this expanded explanation in both the main text and the caption of Figure 3, ensuring readers can fully understand the purpose and implementation of this validation strategy.

- Reviewer comment: In Figure 1 fonts are too small.

Our Reply: We acknowledge the reviewer's comment. The final manuscript will be formatted according to the journal's standards.

- Reviewer comment: In the results section on nuclear envelope analysis, authors state: "We further illustrate SReD's versatility by detecting nuclear envelopes in DAPI- stained cells (11) using empirical reference blocks extracted directly from the input image, distinguishing different morphological states potentially related to cell division or stress (Fig. 1b; Fig. S3)." This is not clear to me, please explain better. A reference to Figure S3d could be more appropriate, since there the differences in the nuclear shapes are more evident and quantified.

Our Reply: Our goal is to demonstrate that SReD can be used to distinguish between different morphological states. In the context of cell nuclei, these states may be related to cell division or stress. To clarify this point, we have included a reference to Note S2 and Figure S3d, which present the corresponding analysis results. These results illustrate how different morphological states can be identified through a quantitative analysis of SReD's output.

- Reviewer comment: In the Methods, section "Relevance mask", the authors state "This is obtained by sampling the variance of the input image using non-overlapping blocks of the same size as those used for the repetition analysis, and then calculating the average of percentile 0.03.". The authors should discuss the reason for the number 0.03.

Our Reply: The choice of the 0.03 percentile threshold for noise estimation represents a carefully optimised compromise in our methodology. This value, derived from the original method's recommendations, balances the need for accurate noise estimation against the risk of including structural information. Using 3% of the blocks with the lowest variance provides sufficient sampling for reliable estimation while minimising the inclusion of image structures that could lead to noise overestimation. Our testing across various datasets confirmed that this threshold consistently yields robust results without requiring adjustment. We have incorporated this rationale into both the Methods section and Note S1 to provide readers with a clear understanding of this critical parameter's selection and its impact on the algorithm's performance.

- Reviewer comment: In the Supplementary Information, Note S1, point 2. Generation of the Relevance Mask, the authors should expand it to be more self-consistent, or at least refence to the Methods. Please define the constant "C" that is found unreferenced in the caption of figure S1.

Our Reply: We added information to Note S1 to make it more self-consistent. We did not reference the Methods section of the main manuscript to avoid sending the reader to a different document. Regarding the constant "C", we added the definition in the caption of Figure S1.

- Reviewer comment: In Note S1, when introducing the SRS the authors should specify that the range is form 0 to 1, of course this is evident from the pictures but should be defined.

Our Reply: We added this information to Note S1.

- Reviewer comment: In Note S2, the authors should put a reference for the "Cosine similarity " function or explain briefly how it works.

Our Reply: We acknowledge the error in referencing the "Cosine similarity" metric, which was inadvertently included due to its use during algorithm development. The correct metric employed in our analyses is the "absolute difference of standard deviations". We have rectified this mistake throughout

the manuscript, replacing all instances of "Cosine similarity" with "absolute difference of standard deviations". To provide clarity, we have included a detailed explanation of this metric in Note S1. Furthermore, we are preparing a new software release that will incorporate the correct formula, ensuring consistency between our manuscript and the publicly available code. This updated version will be published concurrently with the revised manuscript, allowing readers to access and use the accurate implementation.

- Reviewer comment: In Note S3, section "Detection of ring patterns in large fields-of-view", point 3. Generation of angular weight maps, the authors should better explain the weighting procedure, this is not clear at all.

Our Reply: We have expanded the explanation of the angular weight map generation process in Note S3 to enhance clarity. The revised description now details that these maps are derived from the previously calculated angle maps of axon skeletons. To accurately represent the full axon structure rather than just the central line, we apply a series of transformations to the angle maps. Specifically, we employ Gaussian filters to smooth the angle information across the axon width, followed by power functions to accentuate the angular features. This process effectively extends the angular information from the skeleton to encompass the entire axon width. The resulting angular weight maps provide a comprehensive representation of axon orientations throughout the field of view, enabling more precise ring pattern detection across varying axon orientations.

- Reviewer comment: In Note S4, section "Analysis of SReD's robustness against non-specific structures", the detection counts out of the global repetition maps does not show a linear trend, while this is the case of the input image counts. Can the authors comment on this difference?

Our Reply: In the input image counts, the trend appears linear, but we cannot rule out significant differences between samples. For the SReD output, the apparent non-linear trend arises because we plotted the data against the Perlin noise frequency to align all samples to an increasing x-axis. However, if we instead consider the "scale" of the Perlin noise (i.e., the inverse of the frequency) and order the samples accordingly, the data will exhibit a more linear trend. We discussed this while writing the manuscript and decided to plot the frequency instead of the scale to make the plot less confusing.

Reviewer 2:

- Reviewer comment: I co-reviewed this manuscript with one of the reviewers who provided the listed reports. This is part of the NatureCommunications initiative to facilitate training in peer review and to provide appropriate recognition for Early Career Researchers who co-review manuscripts.

Our Reply: We thank Reviewer 2 for taking the time to co-review our manuscript and collaborate in improving its final iteration.

Reviewer 3:

- Reviewer comment: “The analysis proceeds using reference blocks, either simulated or empirically sampled from the image.” It is not explain what “empirically sampled” means.

Our Reply: We have revised our terminology to enhance clarity regarding the reference blocks used in our analysis. "Empirical blocks" refer to those directly extracted from the image data, representing observed structures rather than theoretical constructs. To avoid confusion, we have replaced all instances of "empirically extracted" with "extracted from the image". This change has been consistently applied throughout the manuscript, ensuring a clear distinction between blocks derived from the data itself and those that are simulated. We have also provided additional context in relevant sections to elucidate the process of extracting these blocks from the image, thereby improving the overall comprehension of our methodology.

- Reviewer comment: “These blocks are matched against the input using correlation metrics to generate repetition maps, as detailed in the Methods section.” We think it is better to explain this here in the Results section and not only in the Methods.

Our Reply: We concur with the reviewer's suggestion and have revised this section accordingly. In the updated manuscript, we have incorporated a concise explanation of the block-matching process directly in the Results section. This addition provides readers with an immediate understanding of the core methodology without necessitating reference to the Methods or Supplementary Notes. Specifically, we have included a brief description of how reference blocks are compared to the input image using correlation metrics, and how these comparisons generate the repetition maps.

- Reviewer comment: “We measured an average spacing of 178 nm under control conditions.” How was this measured?

Our Reply: The average spacing of 178 nm in control conditions was determined through autocorrelation analysis of the repetition maps. Specifically, we calculated the autocorrelation function for each extracted region containing periodic patterns. The position of the first harmonic in these autocorrelation functions corresponds to the dominant periodicity in the pattern. By averaging these positions across all analysed regions, we obtained the mean inter-ring spacing of 178 nm. This method provides a robust and unbiased estimation of the periodic structure's spatial frequency, leveraging the enhanced pattern detection capabilities of our SReD algorithm. We have incorporated this detailed explanation into the manuscript to ensure clarity on our quantification approach.

- Reviewer comment: “Nonlinear mapping effectively distinguishes real patterns from imprinted ones.” The explanation of “nonlinear mapping” is missing.

Our Reply: We have expanded our explanation of nonlinear mapping in the revised manuscript. Nonlinear mapping refers to the process of transforming

Structural Repetition Scores (SRSs) using nonlinear functions, typically power functions, to enhance contrast between different structural patterns. This transformation accentuates subtle differences in SRSs, facilitating the visual distinction between genuine structural repetitions and potential artefacts. We have added a detailed description of this process, including its rationale and implementation, to the Methods section. The text now explains how nonlinear mapping enhances the interpretability of repetition maps by amplifying differences between SRSs, thereby improving the detection and characterisation of repetitive biological structures.

- Reviewer comment: “Localisation of round viral structures via local extrema calculation revealed that the repetition map provided a superior platform for extrema detection compared to direct analysis of the raw images.” Please add a reference to a corresponding Figure.

Our Reply: We added a reference to the corresponding Figure (Figure S4).

- Reviewer comment: Clarify interpretation of ref 7,8: As far as we know these methods don't need single molecule localization data, they work with any set of points.

Our Reply: Indeed, the term “single molecule localization data” does not reflect the full extent of data that these methods can process. We changed the wording to “point data”.

- Reviewer comment: In general, for the image data we recommend publication in the BioImageArchive, because it is searchable and will allow to provide a minimal amount of imaging relevant metadata.

Our Reply: We appreciate the reviewer's recommendation and agree that publishing our image data in the BioImageArchive is an excellent suggestion. We followed this advice and deposited our datasets in the BioImageArchive repository ([https://doi.org/ 10.6019/S-BIAD1620](https://doi.org/10.6019/S-BIAD1620)).

- Reviewer comment: For data that is reused from existing publications, we think it is not sufficient to link to the corresponding publication, but the link must directly go to the image data. Otherwise it is not practical for the reader to actually find the data in a realistic amount of time.

Our Reply: We agree and added direct links to the datasets.

- Reviewer comment: We found that the method in general looks very much like a standard template matching approach. Please discuss/compare with other related methods (e.g. cross-correlation based template matching)? For example, how does the proposed method differ from existing implementations such as https://scikit-image.org/docs/stable/auto_examples/features_detection/plot_template.html ?

Our Reply: While SReD incorporates elements of template matching, it extends beyond standard approaches in several key aspects. Unlike Scikit-image's template matching, which is primarily designed for identifying specific patterns, SReD offers a more comprehensive framework for analysing structural repetition across multiple scales.

The core innovation of SReD lies in its "global repetition" mode, which quantifies the relative repetition of all image textures without requiring a predefined template. This unsupervised approach enables the detection of recurring patterns that may not be known a priori, a capability not found in standard template matching libraries. Furthermore, SReD's implementation as an ImageJ/Fiji plugin facilitates seamless integration with existing microscopy workflows, enhancing its practical utility for biologists. The algorithm's flexibility in correlation metrics and its ability to perform multiscale analysis further distinguish it from conventional template matching methods.

In essence, while SReD builds upon established template matching principles, its expanded functionality and tailored design for microscopy applications set it apart as a key approach for structural analysis in biological imaging. To clarify this, we added a Supplementary Table showing a comparison between SReD's functionalities and other methods in the same domain, including Scikit Template Matching.

- Reviewer comment: How is noise defined for the GAT step?

Our Reply: We have expanded our explanation of the Generalised Anscombe Transform (GAT) in the Methods section. The noise variance for the GAT step is estimated using a sampling approach: We calculate the local variance across non-overlapping image blocks and determine the average variance of blocks below the 3rd percentile. This method provides a reliable estimate of the noise floor while minimising the inclusion of structural information. The GAT parameters are then optimised using an error minimisation function, aiming to produce a variance-stabilised image with a noise variance as close to 1 as possible. This process ensures that the GAT effectively addresses the complex noise characteristics typical in microscopy images, including both Poisson and Gaussian components, thereby improving the accuracy of subsequent structural analysis steps.

- Reviewer comment: "and then calculating the average of percentile 0.03." Please rephrase, what do you mean by the average of a percentile?

Our Reply: Indeed, the definition of "percentile" refers to the threshold separating a percentage of the data, and not to the data points below it. We corrected this mistake by changing the text to "average of the data below percentile 0.03".

- Reviewer comment: "This produces a binary mask outlining areas with sufficient structural content." Add link to a Figure.

Our Reply: We added a link to Figure S1, which refers to this section of the manuscript.

- Reviewer comment: “ "1-to-all" or "all-to-all" “: Consider renaming to “Manual Reference” and “Automatic Reference” ?

Our Reply: We appreciate the reviewer's suggestion, but we believe the current terminology more accurately reflects the underlying computational approach. The terms "1-to-all" and "all-to-all" describe the sampling schemes used, which are fundamental to the algorithm's operation regardless of user interaction. The "1-to-all" scheme can be employed both manually and automatically, while the "all-to-all" scheme incorporates multiple automated "1-to-all" analyses. To address the reviewer's concern and improve clarity, we have expanded the explanation in the main text to explicitly link these sampling schemes to their respective operational modes, thereby elucidating the relationship between the sampling methodology and user interaction without altering the core terminology.

- Reviewer comment: “modified cosine similarity”: Please explain the maths.

Our Reply: I acknowledge the reviewer's request for mathematical clarification. The manuscript incorrectly referenced a "Cosine similarity" metric, which was an early experimental measure tested during development but not used in the final analyses. The actual metric employed throughout our studies is the "absolute difference of standard deviations" (ADSD). This metric effectively captures local contrast differences between image regions, providing a robust measure for structural similarity assessment. We have rectified all instances of this terminology in the manuscript and implemented the correct formula in Note S1. The forthcoming software release will incorporate this corrected implementation to ensure consistency between the published methodology and the available computational tools.

- Reviewer comment: “To minimise blocking artefacts, the square blocks are transformed into their inbound elliptical counter-parts”?: Please explain in more detail.

Our Reply: The use of square or rectangular blocks in microscopy image analysis frequently introduces undesirable blocking artefacts that manifest as artificial rectangular patterns in the resulting repetition maps. To address this limitation, SReD implements an elliptical transformation approach for all reference blocks. This process involves inscribing the largest possible ellipse within each rectangular block and masking out all pixels that fall outside this elliptical boundary. The transformation preserves the central structural information while eliminating corner regions that often contribute to edge artifacts. This elliptical masking technique produces smoother transitions between adjacent blocks in the repetition maps and better reflects the natural curvature often found in biological structures. We added this information to the text.

- Reviewer comment: “In the global repetition mode, SR ...”: Is “global repetition mode” the same as “all-to-all”?

Our Reply: The term "all-to-all repetition" refers to a sampling scheme where each image block is compared with all other blocks within the dataset. The Global Repetition mode of our algorithm employs this exact sampling scheme. We acknowledge that this relationship was not explicitly clarified in the original manuscript. To address this, we have revised the text to explicitly state that the Global Repetition mode uses the "all-to-all repetition" scheme, ensuring clarity for readers.

- Reviewer comment: "Non-linear mapping": We wondered whether the non-linear mapping is mathematically similar to the above W weight function? If so, could this be condensed into one step in the algorithm?

Our Reply: The non-linear mapping and the weight function W serve distinct purposes in the SReD algorithm, despite some mathematical similarities. The weight function W , a Gaussian function, is an integral part of the core algorithm, enhancing the contribution of similar blocks to the average Structural Repetition Score (SRS). In contrast, the non-linear mapping is a post-processing step applied to the final repetition map, typically using a power function to enhance visual contrast. While the weight function's parameters are fixed and based on block characteristics and estimated noise variance, the non-linear mapping's parameters can be adjusted by the user to optimise visualisation for different datasets. This separation allows for flexible fine-tuning of the output without recalculating the computationally intensive repetition maps, making it a valuable tool for exploratory analysis of complex biological structures.

- Reviewer comment: "This process was automated using a custom ImageJ macro." Please add a link to the corresponding code repository.

Our Reply: All custom scripts developed for this study are openly accessible in the SReD GitHub repository, as specified in the "Code availability" section of the manuscript. To enhance clarity and avoid including lengthy URLs directly in the main text, we have added a statement explicitly directing readers to the repository for access to these scripts. We updated the text to include a link to the scripts mentioned.

- Reviewer comment: "An image volume containing 3D simulated reference particles was generated using a custom Python script." Please add a link to the corresponding code repository.

Our Reply: Expanding on the previous comment and reply, we added links to the scripts.

- "The repetition maps were non-linearly mapped using a power transformation with an exponent of 1000. NRMSE maps were calculated using the "scikit-image" library (v0.22.0)." Please add a link to the corresponding code repository.

Our Reply: We added the link to the Scikit package used in the "Methods" section to avoid cluttering the main text with long links.

- Reviewer comment: How are borders defined in computing N in equation 5 (“N is the size of the input image excluding borders”)?

Our Reply: The borders are defined as the XY radii of the blocks. We added this information to the text.

- Reviewer comment: Please discuss whether rotated particles would also be detected? A simple cross-correlation would not do this but maybe the authors framework takes rotations into account?

Our Reply: The detection of rotated particles represents a key consideration in SReD's framework design. While traditional cross-correlation approaches struggle with rotated structures due to their inherent sensitivity to pixel positioning, SReD offers a more sophisticated solution through its flexible correlation metrics system. The framework incorporates both rotation-sensitive metrics, such as Pearson correlations, and rotation-invariant metrics like the absolute difference of standard deviations. This dual approach enables users to choose whether rotational sensitivity is desired for their specific application. For instance, when analysing cellular structures that may appear in various orientations, the rotation-invariant metrics prove particularly valuable, allowing detection regardless of the particle's orientation without requiring multiple template rotations. This adaptability in handling rotated structures distinguishes SReD from simpler cross-correlation methods, making it more versatile for diverse biological applications. Furthermore, as demonstrated in Figures 2 and S8, SReD can be used to detect rotated particles using rotated blocks and/or input images. This is particularly important for complex structures, since rotation-invariant metrics often suffer from lower sensitivity to pixel positions.

- Reviewer comment: Please also discuss robustness to: Structures of different sizes? Partial occlusion? Deformations?

Our Reply: The algorithm's sensitivity to structure size is primarily determined by the "block-to-image size" ratio, with larger ratios capturing larger structures and vice versa. This flexibility allows for multiscale analysis, as demonstrated in our examination of Nuclear Pore Complex structures (Figure S5). For partially occluded or deformed structures, SReD assigns lower Structural Repetition Scores (SRSs), effectively distinguishing them from intact structures. Our evaluation of SReD's specificity (Figure S13) further illustrates the algorithm's resilience to these variations. While we believe the current manuscript adequately addresses these aspects, we acknowledge the importance of these considerations in assessing SReD's versatility and reliability across diverse biological contexts.

- Reviewer comment: We did not yet manage to run the software but we are very happy to test a revised version.

Our Reply: Before the software's release, we made efforts to ensure that the installation was possible and straightforward in the most common systems. However, as the reviewer pointed, we did not accomplish this successfully. With the help of the reviewer, who pointed this issue in our GitHub repository, we

managed to resolve the issue. The software is now fully working in Windows, MacOS and Linux machines. Due to the nature of the reviewer's response in the GitHub issues page and their comments regarding software flaws in this revision, we believe the reviewer was then able to use the software and that the issue is solved. For this, we would like to thank the reviewer for pointing the issue and helping in its resolution.

- Reviewer comment: There is a lack of feedback during long computations, e.g. when doing variance stabilisation, nothing visible happens though eventually a new image window pops up. Please add progress reports (ImageJ has a progress bar that could be used).

Our Reply: We appreciate the reviewer's concern regarding the lack of feedback during lengthy computations. In response, we have implemented a comprehensive feedback system in the latest SReD release. This system uses both the ImageJ/Fiji progress bar and the log window to provide users with real-time updates on the analysis progress. The log window now displays messages at each stage of the analysis, informing users of the current step being executed. Concurrently, the progress bar indicates the overall completion percentage. While the progress bar may reset when users interact with the GUI, the combination of these feedback mechanisms ensures that users remain informed throughout the entire analysis process.

- Reviewer comment: As mentioned above, is there support for the analysis of multiple images of the same sample?

Our Reply: SReD currently lacks direct support for analysing multiple images of the same sample through its user interface. However, we recognise the importance of this functionality and plan to incorporate it in our upcoming release. In the interim, users can leverage ImageJ macros to automate SReD analyses across multiple images. This approach involves recording the macro commands for a single analysis using ImageJ's "Record" function, then adapting the resulting script to iterate over multiple images. We recommend this method to circumvent potential system crashes that may occur when queueing numerous analyses simultaneously. This solution offers a practical workaround while we develop a more integrated multi-image analysis feature for future versions of SReD.

Reviewer 4:

- Reviewer comment: In Fig. 3, the authors demonstrate that the program can detect the positions, intensities, and numbers of HIV-1 Gag assemblies in 3D. However, since the virus particles are estimated to be 100-150 nm in diameter, which is below the diffraction limit, recent studies on the virus assemblies have been performed using super-resolution microscopy. therefore, readers would be interested in the ability of SReD to analyze Gag assemblies in super-resolution microscope images, such as the size estimation of assemblies, etc. It would be great to analyse published super resolution microscopy data or STORM data obtained in the authors' system. Alternatively, it would be

interesting and helpful for future users to show similar data using Gag-(i)EGFP with multimerization deficient mutations (e.g. W185A W186A), as the authors demonstrated EGFP intensity of Gag-(i)EGFP as a function of SRS.

Our Reply: In response to the reviewer's suggestion, we have expanded our analysis to address the applicability of SReD in detecting and analysing sub-diffraction limit structures, particularly HIV-1 Gag assemblies, using super-resolution microscopy data. To this end, we incorporated analyses of published STORM datasets and electron microscopy (EM) data to demonstrate SReD's versatility across imaging modalities and its utility in size estimation and structural characterisation.

We have included a new analysis in Note S3 and Figure S6, where SReD was applied to EM data to detect HIV-1 viral particles. This analysis highlights the algorithm's ability to identify and classify viral particles based on their structural features, distinguishing mature virions from other particles. The results were validated against manual annotations and compared with a deep learning-based detection approach, achieving a concordance of 90% for mature particles. These findings underscore SReD's robustness in detecting sub-diffraction structures in EM data.

Additionally, we analysed STORM data of HIV-1 Gag assembly platforms (Note S4 and Figure S7). SReD was used to detect viral assembly platforms in control cells and cells treated with CK666, an actin-debranching drug. The algorithm provided quantitative insights into these platforms' stability and morphological states. Notably, SReD detected a significantly higher number of assembly platforms compared to direct analysis of input images, demonstrating its enhanced sensitivity and utility for super-resolution datasets.

Finally, to facilitate the broader adoption of SReD for similar analyses, we created a video tutorial (Video R1) detailing the processing pipeline for EM and STORM data. This resource guides users through preprocessing steps, parameter optimisation, and result interpretation, ensuring reproducibility and accessibility.

These additions comprehensively address the reviewer's concerns by showcasing SReD's capabilities in analysing sub-diffraction limit structures across multiple imaging modalities while providing a foundation for future studies involving super-resolution microscopy datasets.

- Reviewer comment: In Fig. 4, the authors showed that the EB3 trajectory indicated lower stability, while MTOC indicated higher stability. This conclusion is intuitively known and not surprising when comparing moving spots and static structures, although I recognize SReD's utility in this example, as shown in Fig. S13. I wondered if the program could calculate a EB3 velocity using a combination of the SReD's functions.

Our Reply: The analysis presented in Figure 4 demonstrates SReD's capability to differentiate between dynamic and static cellular structures, such as EB3 comets and the microtubule organising centre (MTOC). While this distinction may seem intuitive, SReD's quantitative approach provides a robust method for objectively assessing structural stability across diverse cellular components. Although SReD does not directly calculate EB3 comet velocities in this particular analysis, it offers a valuable foundation for further quantitative

studies. We conducted experiments using the repetition map as a platform for tracking (Figure R1). Our analysis revealed a modest increase in the number of detected tracks, likely reflecting SReD's ability to enhance structural details. However, we deemed this increase insufficient to warrant a dedicated figure, as SReD's enhancement capabilities were already demonstrated in other figures of our report. We examined various track metrics but found no statistically significant differences between the input data and the repetition map results (Figure R1b). Although the differences were not substantial, we believe it's valuable to share this data with you. We're open to creating a figure if you think this information merits visual representation in our report. Alternatively, we could briefly mention these findings in the text. We appreciate your input on the most appropriate way to present this information, considering that the main property of SReD's structural enhancement has already been established elsewhere in our report.

Figure R1 – **a**) First time frame of the input time lapse (left panel) and the corresponding SReD block repetition map (right panel) calculated using an artificial reference block (inset). The repetition map identifies instances where the reference block, resembling an EB3 comet, recurs. Yellow overlay indicates EB3 trajectories computed with TrackMate; **b**) Kernel density estimate plots depicting the distributions of TrackMate trajectories concerning various variables. Comparisons reveal no significant differences between the original

input data trajectories and those from the SReD repetition maps ($p > 0.05$ for all comparisons; Mann-Whitney U test).

- The Figure legends in Fig. 3 and 4 do not seem to correspond to the figures.

Our Reply: We acknowledge the discrepancy between the figure legends and the corresponding images in Figures 3 and 4. This inconsistency arose from the inadvertent retention of captions from earlier versions of these figures. We have revised both figure legends to ensure they accurately describe the current iterations of Figures 3 and 4.

We highly appreciate the opportunity to address the remaining comments raised by the reviewers. We would like to thank the reviewers for their previous comments that have had a positive impact on our work, helping improve this work. Regarding the remaining concerns, we have included a point-by-point response:

Reviewer #3: *We would like to thank the authors for taking the time to address many of our comments!*

While the manuscript improved in terms of detailed points we still think that the overall structure should be changed.

In this regard we agree with Reviewer 1's comment about the first version of the manuscript: "The reader is scattered from the main text to the supplementary information and to the methods section too many times, making the reading of the article very difficult. The authors may try to put all the comprehensive information on the algorithms in the Supplementary Information, while leaving only the experimental details in the Methods. If not, they should find another way to avoid excessive fragmentation of the information."

Relatedly, we wrote: "In general we found that the main text did not contain enough detail and it was not possible to understand what the method is actually doing. We would recommend to move quite a lot of the methods section into the main text and then have less examples in main text. In fact, some of the examples of running the algorithm on different dataset could be in the supplemental material." (We could not find this section in the rebuttal letter and thus it may have been lost at some point and not accessible to the authors. We are sorry if this was our mistake.)

In our view, this article is about a new analysis method and thus we think this method should be properly described in the main text. There can also be application examples in the main text, but (some of them) could also be in the supplemental material. We think, for a methods paper, there is too little technical information in the main text. We thus still feel that it would be better to rewrite the article such that more technical information is presented in the main text and some of the examples are moved into the supplemental material.

Specifically:

Line 97: "using correlation metrics" does, in our view not contain enough information. As mentioned in our first review, we feel that it should be explained early within the main text how these correlations are computed and what the difference is to simple standard cross-correlation based approaches, which are available in standard libraries, and also as ImageJ plugins such as <https://bmcbioinformatics.biomedcentral.com/articles/10.1186/s12859-020-3363-7>

Specifically, there are different modes that are more or less rotation invariant. It is critical to explain this and at least some of the mathematical foundation to the reader. At line 340 the different correlation metrics are discussed, but since they were not introduced in the main text, one cannot really understand the discussion.

Also when demonstrating the method for different example data, it should be discussed in the text which correlation metrics have been used and why, such that users of the method can make informed decisions. In addition to the correlation metric the other main parameter is the size of the "correlation box"; also here it would be important to add guidelines into the article as to how big this should be chosen and also what happens if one chooses the wrong size.

Relatedly, the "theoretical foundation" section actually starts with implementation details (the fact that this is available as Fiji plugin is, in our view, not a theoretical foundation, the same ideas could be implemented in other platforms). In addition, please explain what a relevance mask is and how it is computed. At the minimum, put a summary of note S1 in the main text and the content of note S1 into the method section. As this is a methods paper, the reader should not need to dig in supplementary docs to get all details.

Our reply: We acknowledge the reviewer's observations concerning the distribution of methodological details within the manuscript. In response and to enhance the clarity and accessibility of our work, we have restructured the article. Critical aspects of the Structural Repetition Detector (SReD) framework, including its theoretical underpinnings and core functionalities, which were previously elaborated in other sections such as the supplementary information, have now been integrated into the main text. Specifically, the main manuscript now presents a detailed account of the operational principles of SReD, starting with applying the Generalised Anscombe Transform for noise variance stabilisation. This is followed by an explanation of the relevance mask generation, which serves to exclude image regions devoid of substantive structural information by quantifying local texture prominence. The procedures for employing reference blocks, whether simulated or empirically derived from the image data, and the subsequent matching against the input image using various correlation metrics are also described. We have clarified the different sampling schemes, such as the "1-to-all" approach for block repetition analysis and the "all-to-all" scheme for unbiased global repetition mapping. Furthermore, the calculation and interpretation of the Structural Repetition Score, including its formulation for both block repetition and global repetition modes, and the application of non-linear mapping for contrast enhancement of repetition maps, are now comprehensively covered within the main body of the paper. This reorganisation now provides readers with a self-contained understanding of SReD's methodology by reading the main manuscript.

Reviewer #3: *Noise variance stabilisation*

This takes several seconds and indication of progress or whether anything is actually being done is important. We thank the authors of implementing progress in the Fiji progress bar, however, we initially did not even realise this, because the progress bar disappears as soon as the mouse is moved. We recommend also indicating progress in the IJ log window. See here how to use the `\\Update` functionality to achieve this: <https://wsr.imagej.net/macros/LogWindowTricks.txt>

The wiki appears to suggest that some (histogram? line profile?) plot should appear along the image but we only see the new image and no plot. Maybe remove the plot from the wiki if it is not critical? In addition, the example in the wiki doesn't show any difference obvious between before and after.

Our reply: We appreciate the suggestion offered by the reviewer. We will incorporate a progress update function in a forthcoming release of the SReD plugin. Regarding the histogram display, as it is not a critical output nor generated by the plugin itself, we have removed it from the associated wiki documentation.

Reviewer #3: *Relevance mask*

We found this to be rather slow (even though it was using the GPU): Is it supposed to take >30s to compute for a 2000x2000 image?

Our reply: We acknowledge the reviewer's comment regarding the execution speed of the relevance mask calculation. We concur that the current processing time for this step can be considerable. The generation of the relevance mask necessitates a comprehensive analysis of local image texture across the entirety of the input data. This process involves the systematic interrogation of numerous image blocks, for each of which statistical parameters such as local variance are computed to quantify texture prominence. The aggregation of these individual, block-based calculations, particularly for large image datasets, inherently contributes to the observed computational time. Nonetheless, we are committed to enhancing the user experience and will actively explore code optimisation strategies aimed at accelerating this calculation and reducing its overall execution time in future updates to the SReD plugin.

Reviewer #3: *Choosing the box size and ROIs*

The requirement for odd numbers is cumbersome, especially when drawing a ROI to select the reference mask. We recommend to automatically select the closest odd number and display a warning about it in

the log window. As it is, when one makes a mistake, one needs to go back, find the plugin in the menu and start again.

Our reply: We thank the reviewer for this suggestion and will implement it in a future release of the plugin.

Reviewer #4: *The utility of SReD in detecting sub-diffraction size particles, such as HIV-1 Gag assembly and virus, is well demonstrated by these new supplementary figures with different image resolutions. In new Figs. S6 (b and c) and S7a, side-by-side presentations of treated input images and the block repetition maps, would enhance clarity about how the analysis works. Figure R1 shows that SReD can output sufficient repetition maps to allow downstream analysis of the EB3 trajectory in the tracking program TrackMate, demonstrating another application of SReD data. The authors may consider including this figure in the supplementary materials and explanation in the text and the Methods section (or Supplementary Note). Furthermore, If SReD offers advantages over directly using other tracking programs, incorporating these benefits would be valuable for potential users.*

Our reply: We are grateful to the reviewer for their comment and for highlighting, with Figure R1, the prospective versatility of SReD's outputs for subsequent analyses. Regarding its inclusion in the present manuscript, our primary aim is to introduce the SReD methodology, thoroughly validate its core functionalities, and demonstrate its application in diverse structural repetition detection scenarios. While the integration with tracking software is a valuable application, a detailed exploration, particularly one that would rigorously assess any advantages SReD might offer over directly using other tracking programs, would require a dedicated study with specific experimental designs and comprehensive comparative validation. Such an investigation, while certainly worthwhile, extends beyond the immediate focus of this paper on SReD.